# Human DNA polymerase delta is a pentameric holoenzyme with a dimeric p12 subunit

Prashant Khandagale, Doureradjou Peroumal, Kodavati Manohar, Narottam Acharya ⓘ

**Human DNA polymerase delta (Polδ), a holoenzyme consisting of p125, p50, p68, and p12 subunits, plays an essential role in DNA replication, repair, and recombination. Herein, using multiple physicochemical and cellular approaches, we found that the p12 protein forms a dimer in solution. In vitro reconstitution and pull down of cellular Polδ by tagged p12 substantiate the pentameric nature of this critical holoenzyme. Furthermore, a consensus proliferating nuclear antigen (PCNA) interaction protein motif at the extreme carboxyl-terminal tail and a homodimerization domain at the amino terminus of the p12 subunit were identified. Mutational analyses of these motifs in p12 suggest that dimerization facilitates p12 binding to the interdomain connecting loop of PCNA. In addition, we observed that oligomerization of the smallest subunit of Polδ is evolutionarily conserved as Cdm1 of *Schizosaccharomyces pombe* also dimerizes. Thus, we suggest that human Polδ is a pentameric complex with a dimeric p12 subunit, and discuss implications of p12 dimerization in enzyme architecture and PCNA interaction during DNA replication.**

## Introduction

Accurate and processive DNA synthesis by DNA polymerases (Pol) during chromosomal DNA replication is essential for lowering the rate of spontaneous mutations and suppressing carcinogenesis (Pavlov et al, 2006). Three essential DNA polymerases, namely, Polα, Polδ, and Polε, coordinate eukaryotic chromosomal DNA replication (Stillman, 2008; Kunkel & Burgers, 2014, 2017; Burgers & Kunkel, 2017). Based on biochemical and genetic studies, mostly those carried out in the budding yeast, it has been proposed that Polα initiates DNA replication by synthesizing a short RNA–DNA primer, and is followed by loading of DNA clamp proliferating cell nuclear antigen (PCNA) by its loader replication factor C. Polδ plays a major role in synthesizing "Okazaki fragments" in the lagging and initiating leading-strand DNA synthesis (Aria & Yeeles, 2018). Polε is involved in only leading-strand DNA synthesis (Acharya et al, 2011; Johnson et al, 2015). In the absence of Polε, Polδ also synthesizes the bulk of the leading strand. The mechanism of DNA replication in higher eukaryotes is yet to be deciphered; however, Polδ replicates both the leading and lagging strands of the SV40 virus genome (Waga et al, 1994; Stillman, 2008). Irrespective of their different roles in DNA replication, these DNA polymerases possess certain commonalities such as the multi-subunit composition and signature sequences of a B-family DNA polymerase in the largest catalytic subunits (Tahirov et al, 2009; Kunkel & Burgers, 2017).

Among the replicative DNA polymerases, the subunit composition of Polδ varies between eukaryotes. Whereas *Saccharomyces cerevisiae* Polδ consists of three subunits, Pol3, Pol31, and Pol32, Polδ from *Schizosaccharomyces pombe* possesses four subunits, Pol3, Cdc1, Cdc27, and Cdm1 (Zuo et al, 2000; Acharya et al, 2011; Miyabe et al, 2011). The mammalian Polδ holoenzyme consists of p125 as the catalytic subunit, the yeast homologue of Pol3, whereas p50, p68, and p12 are the structural subunits (Zhou et al, 2012). The accessory subunits p50 and p68 are the equivalents of Pol31/Cdc1 and Pol32/Cdc27 subunits, respectively. The p50/Pol31/Cdc1 subunit makes a connecting bridge between the catalytic subunit p125/Pol3 and p68/Pol32/Cdc27 and is indispensable for cell viability. Although Pol32 is not essential for cell survival in *S. cerevisiae*, in its absence, cells exhibit sensitivity to both high and cold temperatures, and susceptibility to genotoxic stress (Johansson et al, 2004). Contrary to this, Cdc27 deletion strain of *S. pombe* is not viable (Bermudez et al, 2002). The nonessential p12 subunit is the Cdm1 homologue and is absent in *S. cerevisiae*. Yeast two-hybrid and co-immunoprecipitation analyses suggested a dual interaction of p12 with p125 and p50; however, the modes of binding among these subunits are yet to be defined (Li et al, 2006). In vitro reconstitution has facilitated purification of four different subassemblies of human Polδ (hPolδ), such as p125 alone, p125-p50 (core complex), p125-p50-p68, and p125-p50-p68-p12 complexes, for biochemical studies. Reports also suggest that the subunit composition of hPolδ may alter in vivo with cellular response to DNA damage (Lee et al, 2012, 2014). Upon treatment of human cells with genotoxins such as UV, methyl methanesulfonate, hydroxyurea, and aphidicolin, the p12 subunit undergoes rapid degradation to result in a trimeric hPolδ (p125/p50/p68) equivalent to ScPolδ with higher proofreading activity (Meng et al, 2010). Thus, p12 subunit seems to play a crucial role in regulating Polδ function.

Laboratory of Genomic Instability and Diseases, Department of Infectious Disease Biology, Institute of Life Sciences, Bhubaneswar, India

Correspondence: narottam_acharya@ils.res.in

The function of Polδ as a processive DNA polymerase mostly depends upon its association with PCNA that acts as a sliding clamp (Krishna et al, 1994). The interaction of PCNA-binding proteins with PCNA gets mediated by a conserved PCNA-interacting protein motif (PIP-box) with a consensus sequence Q-x-x-(M/L/I)-x-x-FF-(YY/LY), where x could be any amino acid (Haracska et al, 2005; Yoon et al, 2014). Previously, we have shown that all the three subunits of ScPolδ functionally interact with trimeric PCNA, an interaction mediated by their PIP motifs (Acharya et al, 2011). All three PIP boxes of ScPolδ are required to achieve higher processivity in vitro. Similarly, reports from hPolδ studies suggest that all the four subunits of hPolδ are involved in a multivalent interaction with PCNA and each of them regulates processive DNA synthesis by Polδ (Wang et al, 2011). The PIP motifs have been identified in p68 and p50, but the same is yet to be mapped in p125 and p12 (Zhang et al, 1999; Lu et al, 2002). Studies based on far-Western and immuno-precipitation analyses revealed that the first 19 aa of p12 are involved in PCNA interaction, although the stretch lacks the canonical PIP box sequence (Li et al, 2006; Terai et al, 2013).

In this study, we have reinvestigated the interaction of p12 with other Polδ subunits and PCNA. Our results indicate that the smallest subunit p12 exists as a dimer in solution to establish a dual interaction with both p125 and p50 subunits of Polδ. We have mapped the dimerization motif to the amino-terminal end and have identified a novel conserved PIP box in the C-terminal tail of p12. Importantly, the dimerization of p12 facilitates its interaction with PCNA. Based on our observations, we propose that hPolδ exists in a pentameric form in the cell in addition to other subassemblies, and we discuss the effect of p12 dimerization on PCNA binding with the various subassemblies.

## Results

### Oligomerization of p12 subunit of human Polδ

In several studies, hPolδ holoenzyme has been purified from either an insect cell line or bacterial expression system by using standard chromatography techniques for biochemical characterizations (Fazlieva et al, 2009; Rahmeh et al, 2012; Zhou et al, 2012). Although the purified Polδ used in the various enzymatic assays contained all the four subunits p125, p50, p68, and p12, in most cases, the subunits were not in equimolar concentration. Especially, the smallest subunit p12 was comparatively in higher stoichiometry as compared with others (Fazlieva et al, 2009; Rahmeh et al, 2012). Even when we attempted to purify the hPolδ holoenzyme by using GST-affinity beads, the band intensity of p12 protein was consistently higher than that of the other subunits (Fig S1). Such discrepancy in the composition of Polδ could arise because of either the oligomeric status of p12 or multi-subunit p12 interaction with p125 and p50 or because of staining artifacts. Yeast two-hybrid assay and native PAGE analysis were carried out to examine the potential oligomerization status of the p12 subunit (Fig 1). p12 orf was fused in frame with both GAL4 activation (AD) and GAL4-binding domains (BD). Other hPolδ subunit orfs were fused with the GAL4-binding domain alone. The HFY7C yeast reporter strain harboring the AD-p12

plasmid was transformed with one of the BD-Polδ subunit plasmids and selected on a Leu⁻Trp⁻ SDA plate. The interactions of p12 with other subunits of Polδ in these transformants were analyzed by selecting them on plates lacking histidine. Growth on a His⁻ plate demonstrates the interaction between the two fusion proteins as only the binding of two proteins makes it possible to form an intact GAL4 activator to confer HIS expression. As reported earlier, p12 interaction was observed with p125 and p50 but not with p68 (Fig 1A, sectors 1, 2, and 3) (Li et al, 2006). Surprisingly, the p12 subunit interacted with itself to give HIS expression (sector 4), whereas no growth was observed in the negative controls (sector 5 and 6). This result suggests that p12 makes specific multivalent interaction with itself and p125 and p50 subunits, but not with p68.

Furthermore, to ascertain oligomerization of p12, the protein was purified to near homogeneity from bacterial cells by using GST-affinity column chromatography, GST-tag was cleaved off by PreScission protease, and analyzed by both native and SDS-containing PAGE. The predicted MW of p12 is ~12 kD with a pI of 6.3. p12 is known to possess abnormal migration in SDS–PAGE (Podust et al, 2002), and similarly, we observed the protein to migrate at ~15-kD molecular weight size position (Fig 1B, lane 2). By taking advantage of native PAGE analysis where proteins resolve based on their charge and hydrodynamic size, we found that p12 migrated at a similar position with Carbonic anhydrase (CA) which is a protein of 30 kD with pI 6.4 (lane 4 and 5). Thus, the similar migration of two proteins in non-denaturing PAGE indicates that both CA and p12 possess similar mass to charge ratio and the slower migration of p12 indicates that it is potentially a dimeric complex.

By taking advantage of isothermal calorimetry (ITC) technique, the oligomerization of p12 protein was examined (Fig 1C, i). p12 was placed both in the sample cell and in the syringe of the calorimeter. Twenty times 2 μl of p12 was injected to p12-containing cell, and the binding of the ligand to the protein was analyzed by monitoring the change in heat. Upon p12–p12 interaction, the ΔH, ΔG, and Kd for the complex were estimated as −1.82 kcal/mol, −9.48 kcal/mol, and ~146 nM, respectively. The number of ligand-binding site as derived from the ITC analysis was found to be ~0.6, which is ~1:1 binding of p12 monomers. Thus, both in vivo and physiochemical studies indicate homodimerization of the p12 subunit.

### Cellular existence of oligomeric p12 in Polδ complex and it's co-localization with PCNA

HEK293 cells were transfected with GFP-p12 construct to establish p12 oligomerization in its cellular state. Such cells will harbor both native and GFP-tagged p12, and co-immunoprecipitation of Polδ from such cell lysate will facilitate easy detection of oligomeric p12 in the complex. Native Polδ holoenzyme was immunoprecipitated by using either anti-GFP (i) or anti-p125 (ii) antibody (Fig 2A). Each of the four native subunits (p125, p68, p50, and p12) was detected by probing with subunit-specific antibody, whereas an anti-GFP antibody detected GFP-p12. While GFP-p12 pulled down cellular Polδ with native p12, anti-p125 antibody precipitated Polδ complex with both the forms of p12 (native and GFP-tagged). In both of the pull-down assays, irrespective of the antibody used, we detected the presence of five subunits of Polδ in the beads. So, p12 indeed

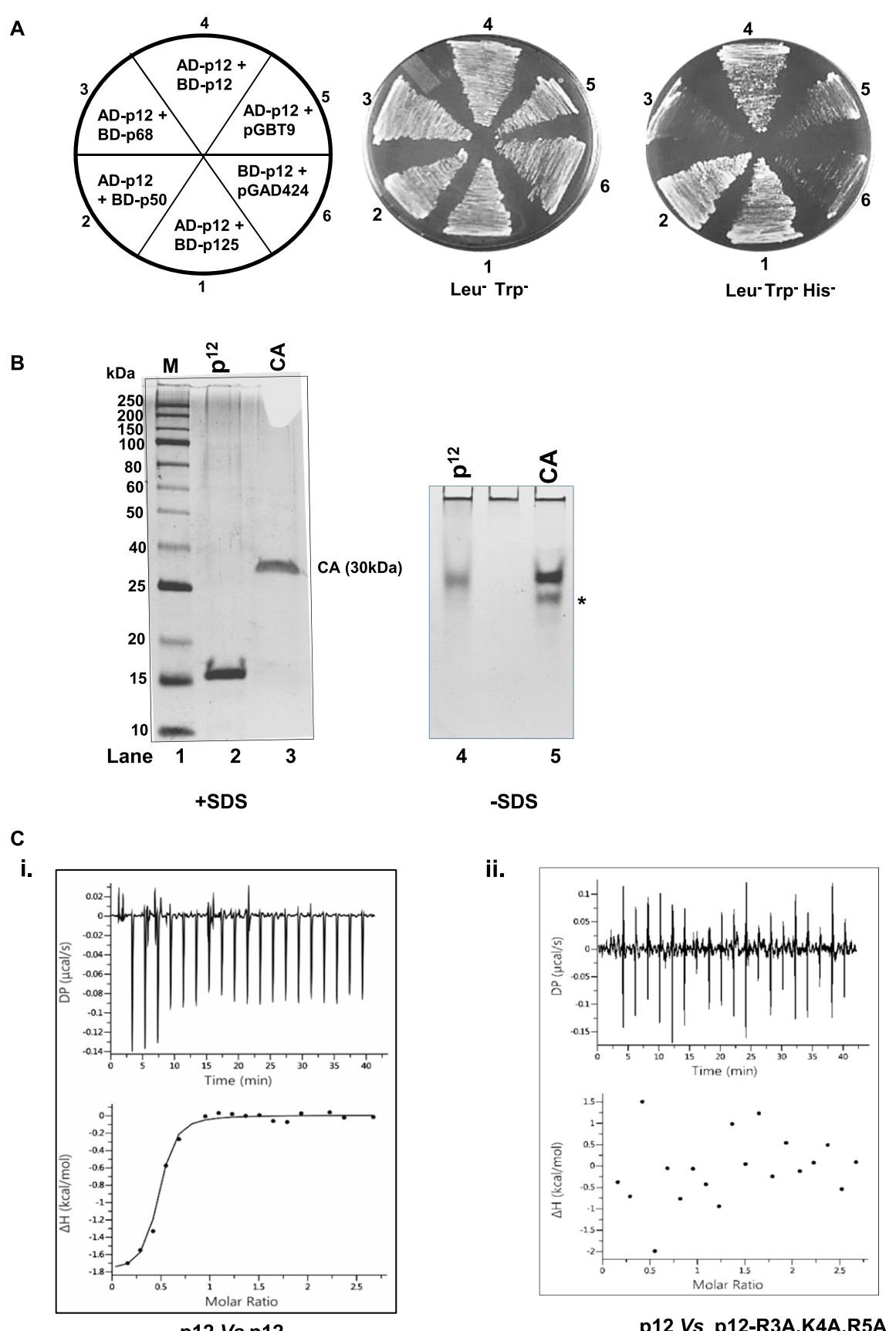

**A**

Disk diagram with sectors:
- 4: AD-p12 + BD-p12
- 5: AD-p12 + pGBT9
- 6: BD-p12 + pGAD424
- 1: AD-p12 + BD-p125
- 2: AD-p12 + BD-p50
- 3: AD-p12 + BD-p68

Leu⁻ Trp⁻

Leu⁻ Trp⁻ His⁻

**B**

CA (30kDa)

Lane 1 2 3 4 5

+SDS          −SDS

**C**

**i.** p12 *Vs* p12

**ii.** p12 *Vs* p12-R3A,K4A,R5A

exists as an oligomer in the Polδ complex even in the cellular context.

PCNA that functions as a cofactor for DNA polymerases orchestrates the replisome by recruiting multiple proteins involved in DNA transaction processes. PCNA forms distinct foci or replication factories indicative of active DNA replication entities within the nucleus (Essers et al, 2005). The p68 and p50 subunits of hPolδ were shown to form foci and co-localized with PCNA in several cell lines (Ducoux et al, 2001; Pohler et al, 2005). The CHO cells were transfected with GFP-PCNA or GFP-p12 and RFP-p12 fusion constructs to examine the physiological relevance of p12 oligomerization and PCNA interaction. As shown in Fig 2B, irrespective of GFP (stained green) or RFP (stained red) fusion, p12 formed discrete compact foci. The subsequent merging of foci in co-transfectant of GFP-p12 and RFP-p12 resulted in the appearance of yellow foci (i). Thus, 100% coincidental accumulation of both the p12 foci suggests that both the proteins are part of a replication unit and function together. Similarly, we have also observed subcellular co-localization of p12 with PCNA as yellow foci appeared by merging foci of GFP-PCNA and RFP-p12 (ii). However, in a similar condition, we did not notice any co-localization of GFP-p12 and RFP-Polθ foci (iii), whereas Polθ co-localizes with PCNA (our unpublished observation). We suggest from these observations that p12 could function in replication factories as a potential oligomeric protein of Polδ with PCNA.

## Identification of motifs involved in p12 dimerization and interaction with PCNA

The primary sequences of p12 from human, mouse, bovine, and *S. pombe* were aligned to identify dimerization and PIP motifs in p12 (Fig 3A). The CLUSTAL W alignment analysis showed a high degree of amino acid conservation of p12 sequences in mammals, and they showed only ~17% identity with *S. pombe* orthologue Cdm1. The carboxyl termini of these proteins displayed better conservation than the amino termini. Cdm1 is composed of 160 aa, and the divergence is apparently due to possession of an insert of about 38 aa exactly in the middle of the p12 homologous sequences in Cdm1. We thought to examine the role of the two highly conserved sequences located at the extreme ends of p12: a basic tripeptide sequence $_3RKR_5$ motif (henceforth referred as RKR motif) and a putative PIP box sequence $_{98}QCSLWHLY_{105}$ (henceforth referred as PIP motif). The putative PIP motif is located in the carboxyl-terminal tail of p12, the usual position of the PIP motif in most of the DNA polymerases, and appears to be very similar to known PIP sequences (Fig 3B). Because an earlier study reported $_4KRLITDSY_{11}$ (a part of RKR motif) as a PCNA-binding region of p12 (Li et al, 2006), we wanted to compare the model structure of this peptide with

$_{98}QCSLWHLY_{105}$. Structurally, PIP box sequences are highly conserved, and formation of a $3_{10}$ helix is a characteristic feature of such sequences. The amino acid stretch encompassing RKR (1-MGRKRLITDSYPVK-14) and PIP (92-GDPRFQCSLWHLYPL-106) domains was used for peptide structure prediction by using PEP-FOLD3 server (http://bioserv.rpbs.univ-paris-diderot.fr/services/PEP-FOLD3/) rather than using a known template-based prediction to avoid any bias. Furthermore, the models were validated by the SAVES and Ramachandran plot (Fig S2A and B), which showed most of the residues in allowed regions. Our structural prediction suggested that the first 10 aa of the RKR motif form an α-helix, whereas the PIP motif of p12 forms a typical $3_{10}$ helix, the structure that fits snugly into the interdomain connecting loop (IDCL) domain of PCNA. The p12 PIP structure was further aligned with available X-ray crystal structures of PIP peptides from p21 (1AXC) and p68 (1U76). The superimposition shows a high degree of similarity between the PIP motifs (Fig 3C). Similarly, the p12 peptide structure was aligned with that of the p68 PIP-hPCNA co-crystal structure, and a remarkable overlapping between the structures was observed (Fig 3C). Thus, our in silico analysis indicated that the C-terminal PIP motif $_{98}QCSLWHLY_{105}$ is the most probable motif that interacts with PCNA other than the N-terminal RKR motif $_4KRLITDSY_{11}$.

Two p12 mutants were generated by mutating residues R3, K4, R5 and L104, Y105 to alanines; and their interaction with wild-type p12 and PCNA was analyzed by yeast two-hybrid approach for providing experimental evidence to our in silico prediction (Fig 3D). As depicted, whereas transformants of BD-p12 with AD-PCNA or AD-p12 grew on SDA plate lacking leucine, tryptophan, and histidine amino acids (rows 5 and 6), both R3A, K4A, R5A and L104A, Y105A p12 mutants were unsuccessful in interacting with PCNA, and thus, no growth was observed (rows 2 and 4), including the vector control (rows 7 and 12). Interestingly, R3A, K4A, R5A mutant is also defective in p12 interaction in yeast cells but not the L104A, Y105A mutant (compare row 1 with row 3). These in vivo results suggest that whereas the RKR motif plays a dual role in dimerization and PCNA interaction, $_{98}QCSLWHLY_{105}$ is involved only in PCNA binding.

Furthermore, we wanted to examine whether dimerization of p12 is also required for other Polδ subunit interactions. Yeast two-hybrid analyses of co-transformants harboring AD-R3A, K4A, R5A with BD-p125 or BD-p50 demonstrated that mutation in this motif does not affect Polδ subunit interaction, as transformants grew efficiently on the Leu- Trp- His- plate (Fig 3D, sectors 8–11). Thus, dimerization of p12 is not required for p125 or p50 binding. However, it is not clear whether p125 and p50 bind to the same or different regions in p12. Nonetheless, dimerization could facilitate binding of p125 and p50 subunits of Polδ to separate monomer of the p12 dimer.

---

**Figure 1.   Interaction of p12 with hPolδ subunits.**
**(A)** Yeast two-hybrid analysis showing the interaction of p12 with the various subunits of hPolδ. HFY7C yeast transformants with various GAL4-AD and GAL4-BD fusions were selected on SD media plates lacking leucine and tryptophan, and with and without histidine amino acid. Sector 1, AD-p12 + BD-p125; Sector 2, AD-p12 + BD-p50; Sector 3, AD-p12 + BD-p68; Sector 4, AD-p12 + BD-p12; Sector 5, AD-p12 + pGBT9; and Sector 6, AD-p12 + pGAD424. **(B)** The purified p12 protein was resolved in native and SDS–PAGE gels. Lane 1: MW; lanes 2 and 4: p12; and lanes 3 and 5: CA. * indicates degraded CA protein. **(C)** ITC analysis of p12 to wild-type (i) or RKR-mutant p12 (ii). In each panel, the upper half shows the measured heat exchanges during each protein injection. The lower half of each panel shows the enthalpic changes as a function of the molar ratio of p12 to wild-type or RKR-mutant p12 monomer. Circles and lines denote the raw measurements and the fitting to a one set of identical sites. Source data are available for this figure.

## A

### i. IP using anti-GFP-p12    ii. IP using anti -p125

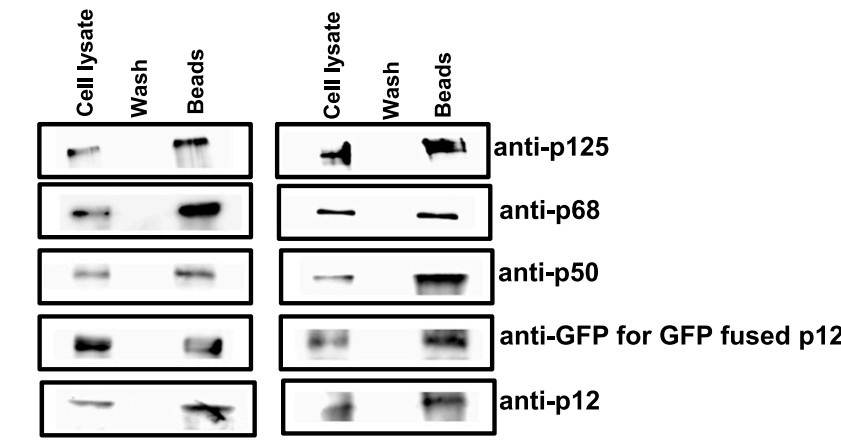

**Figure 2.    Existence of p12 oligomers in the cellular context of Polδ.**
**(A)** Native hPolδ was co-immunoprecipitated from HEK293 cells transfected with GFP-p12. Either anti-GFP (i) or anti-p125 (ii) antibody was used to immunoprecipitate cellular Polδ. After thorough washings, the eluate was separated in 12% SDS–PAGE, and the presence of various subunits of hPolδ was detected by the subunit-specific antibody. GFP-p12 was detected by the anti-GFP antibody. **(B)** Nuclear co-localization of p12 and PCNA. CHO cells were co-transfected with GFP-p12 and RFP-p12 (i), GFP-PCNA and RFP-p12 (ii), or GFP-p12 and RFP-Polθ (iii). After 48 h, the cells were fixed and mounted as described in the Materials and Methods section, and images were taken using Leica TCS SP5 at 63× objective. Scale bar is equal to 5 μm.
Source data are available for this figure.

## B

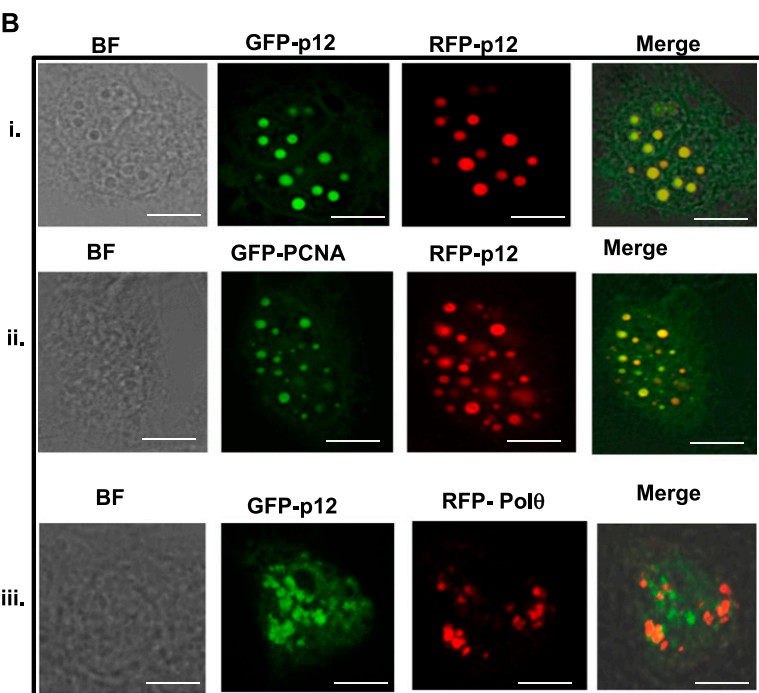

### RKR motif of p12 is critical for dimerization

Multiple basic amino acid motifs such as RKR/KKR/KRK in other proteins are known to play crucial roles in a variety of cellular processes such as their retention and exit from ER, and nuclear localization, as well as the gating of $K^+$ channels (Joiner et al, 1997; Zerangue et al, 1999; Fagerlund et al, 2002; Jones et al, 2007). Such motifs are also found to be involved in protein–protein interaction (van Hennik et al, 2003; Wang et al, 2012; Collins et al, 2014). The mutant proteins R3A, K4A, R5A and L104A, Y105A were purified to near homogeneity and analyzed in SDS–PAGE to find out the involvement of the RKR motif in p12–p12 dimerization (Fig S3, lanes 3 and 4). The wild-type and mutant p12 proteins resolved at a similar position in denaturing PAGE. Thus, mutations in these residues had

no obvious effect on protein mobility and stability. When the proteins were further resolved in non-denaturing PAGE, both wild-type, and L104A, Y105A proteins co-migrated; and their migration was similar to that of CA, as shown earlier (Fig 4A, compare lane 1 with lanes 2 and 4). However, R3A, K4A, R5A mutant p12 was migrating much faster than the other two p12 proteins in the native gel, as a monomer (Fig 4B compare lane 3 with lanes 2 and 4). Therefore, we concluded that the RKR motif is required for dimerization, and mutations in the $_{98}$QCSLWHLY$_{105}$ motif did not impede dimerization.

Furthermore, we compared the gel filtration elution profiles of wild-type and R3A, K4A, R5A mutant p12 by separating an equal amount of proteins through $S_{200}$ molecular exclusion chromatography at physiological salt concentration. Whereas the

**A**

Dimerization motif

```
M. musculus   p12   (1-32 aa)   MGRKRF-----------ITDSYPVVKKREGPPGHSKGELAPEL--------------------
B. taurus     p12   (1-32 aa)   MGRKRL-----------ITDSYPVVKRREGSAGHSKGELAPDL--------------------
H. sapiens    p12   (1-32 aa)   MGRKRL-----------ITDSYPVVKRREGPAGHSKGELAPEL--------------------
S. pombe      Cdm1  (1-60 aa)   M-KRTTQAKKSGQNTNIRDVFPHVVRSNSSQSHIGKKVSSEQSPTPDVTITTKTLDERIKEDD
                                * :**.           * * : :.  .*    ::: :
```

```
M. musculus   p12   (33-72 aa)   ------------------GEDTQSLSQEETELELLRQFDLAWQYGPCTGITRLQRWSRAEQ
B. taurus     p12   (33-72 aa)   ------------------GEEPLPLSVDEEELELLRQFDLAWQYGPCTGITRLQRWHRAEQ
H. sapiens    p12   (33-72 aa)   ------------------GEEPQPRDEEEAELELLRQFDLAWQYGPCTGITRLQRWCRAKQ
S. pombe      Cdm1  (61-120 aa)  ELSKEVEEAWNQIMAERISEPIHCENITKVEFILHHFDTTARYGPYLGMTRMQRWKRAKN
                                                     .:     .     :*::** : :***  *:**:*** **::
```

PIP

```
M. musculus   p12   (73-107 aa)  MGLKPPLEVYQVLKAHPEDPHFQCSLWHLYPL----- 107
B. taurus     p12   (73-107 aa)  MGLKPPPEVHQVLQSHPGDPRFQCSLWHFYPL----- 107
H. sapiens    p12   (73-107 aa)  MGLEPPPEVWQVLKTHPGDPRFQCSLWHLYPL----- 107
S. pombe      Cdm1  (121-160 aa) FNLNPPETVGKILMLEEADEENRKRESLFYDLQTIPG 160
                                 :.*:** * ::*  . * . :      :* *
```

**B**

```
ScPolδ- Pol3  995  KGGLMSFI 1003
ScPolδ- Pol31 321  DKSLESYF 328
ScPolδ- Pol32 331  QGTLESFF 347
hPolδ-  p68   456  QVSITGFF 463
hPolδ-  p50   57   LIQMRPFL 64
hPolδ-  P12   98   QCSLWHLY 105
hPolδ-  p12   4    KRLITDSY 11
ScRad30-pip   618  SKNILSFF 625
hRad30 pip2   437  STDITSFL 444
hRad30 pip1   701  MQTLESFF 708
       Consensus   QXXLXXFF
```

**C**

IDCL of PCNA monomer

$3_{10}$ helix

$3_{10}$ helix

**D**

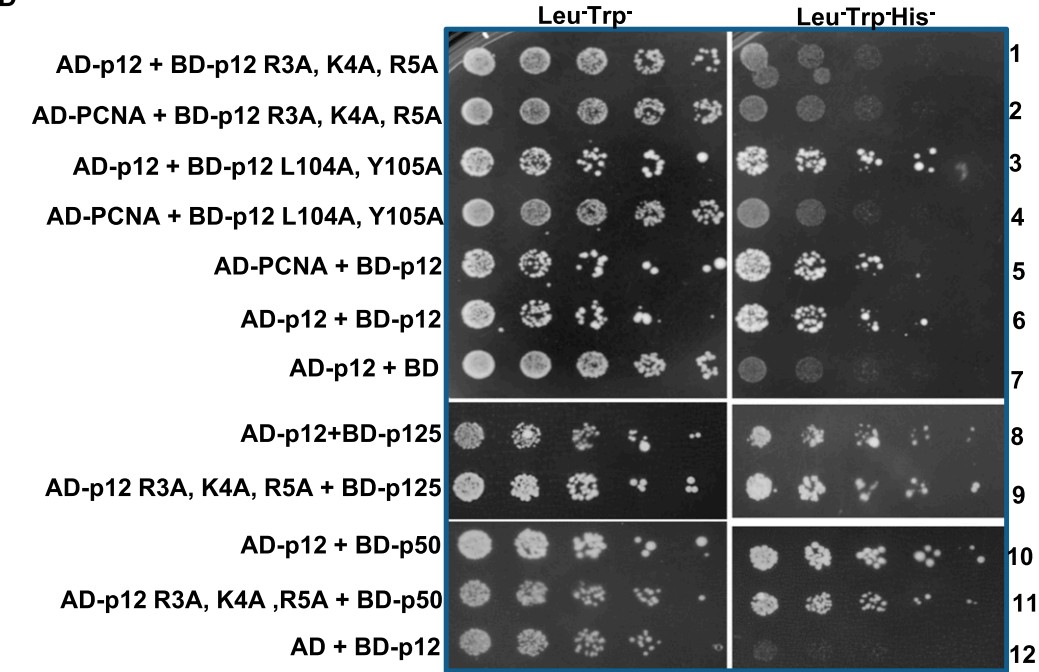

**Figure 4. RKR motif is involved in dimerization.**
**(A)** R3A, K4A, R5A and L104A, Y105A p12 proteins were resolved in native PAGE. Lane 1: CA; lane 2: p12; lane 3: R3A, K4A, R5A; and lane 4: L104A, Y105A. * represents degraded CA. **(B)** About 10 μg of wild-type (red line) and R3A, K4A, R5A (grey line) p12 proteins were subjected to size-exclusion chromatography. Two elution peaks at ~1.4 and ~2.2 ml were observed representing dimer and monomer populations, respectively. **(C)** Immunoprecipitation of GFP-p12 by FLAG-p12 (lane 3) but not of GFP-R3A, K4A, R5A mutant (lane 5). Lane 1: cell lysate input; lanes 2 and 4: washings from beads. Source data are available for this figure.

wild-type p12 protein eluted in two peaks of volume at ~1.4 and ~2.2 ml, corresponding to an oligomeric and monomeric state of the protein (red line), R3A, K4A, R5A mutant eluted (grey line) as a single peak at a volume of ~2.2 ml (Fig 4B). This demonstrates the dimerization of p12 mediated by the RKR motif. This result also rules out any change in stokes radius and residual charge of p12 protein causing abnormal migration in native PAGE due to mutations in the dimerization motif. Co-immunoprecipitation experiment was also carried out in the cell lysates harboring

FLAG-p12 and GFP-p12 or GFP-p12 R3A, K4A, R5A by using anti-FLAG antibody-conjugated beads; and further probed with an anti-GFP antibody (Fig 4C). Although anti-FLAG antibody could pull down wild-type p12 as detected by the anti-GFP antibody, it did not precipitate the RKR mutant (compare lanes 3 and 5). Our yeast two-hybrid, native PAGE, size exclusion chromatography, and pull-down assays demonstrated that indeed RKR motif is a protein–protein interaction motif, and in this study, we show that it is essential for p12 dimerization.

**Figure 3. Identification of p12 motifs involved in dimerization and PCNA interaction.**
**(A)** The primary sequences of the fourth subunit of Polδs from human (*Homo sapiens*), mouse (*Mus musculus*), bovine (*Bos taurus*), and fission yeast (*S. pombe*) were aligned by CLUSTALW. Conserved dimerization motif and PIP-box sequences are colored brown. Identical residues are marked with *, conserved residues are marked with :, and less conserved residues are marked with ·. **(B)** Identified PIP motif residues from DNA polymerase η and δ subunits were aligned and compared with putative PIP sequences in p12. Repeated residues in more than one sequences are highlighted. **(C)** Superimposition of model $3_{10}$ helix structure of p12 PIP (cyan) with available PIP structures from p21 (orange) and p68 (red) peptides without and with human PCNA monomer. **(D)** Yeast two-hybrid analysis showing the interaction of mutants p12 with wild-type p12 and PCNA. HFY7C yeast transformants with various GAL4-AD and GAL4-BD fusions were selected on SD media plates lacking leucine and tryptophan, and with and without histidine amino acids. Row 1: AD-p12 + BD-p12 R3A, K4A, R5A; row 2: AD-PCNA + BD-p12 R3A, K4A, R5A; row 3: AD-p12 + BD-p12 L104A, Y105A; row 4: AD-PCNA + BD-p12 L104A, Y105A; row 5: AD-PCNA + BD-p12; row 6: AD-p12 + BD-p12; row 7: AD-p12 + pGBT9; row 8: AD-p12 + BD-p125; row 9: AD-p12 R3A, K4A, R5A + BD-p125; row 10: AD-p12 + BD-p50; row 11: AD-p12 R3A, K4A, R5A + BD-p50; and row 12: AD-p12 + pGBT9. Source data are available for this figure.

### Formaldehyde cross-linking reveals dimerization of p12

Our native PAGE and ITC analyses suggested potential dimeric nature of p12 as it was migrating at a similar position with CA (30 kD). To estimate the exact number of p12 molecules in the oligomeric complex, we used a formaldehyde cross-linking assay. Reagents such as formaldehyde or glutaraldehyde cross-links neighboring lysine or arginine residues of proteins to form a stable complex that can even withstand SDS denaturation (Manohar & Acharya, 2015); moreover, our mutational analysis deciphered the involvement of the RKR motif in dimerization. Therefore, purified recombinant proteins were cross-linked with formaldehyde, analyzed on 12% SDS–PAGE, and detected by Coomassie brilliant blue staining (Fig 5A). Upon treatment with the cross-linker, both wild-type and L104A, Y105A mutant p12 proteins showed concentration-dependent cross-linked dimers (lanes 2, 3, 4, 10, 11, and 12) that were migrating below 32 kD position and did not form any higher order oligomers, whereas R3A, K4A, R5A mutant protein remained as monomer (lanes 6, 7, and 8). Without any cross-linker, all the three

proteins migrated to the bottom of the gel (lanes 5, 9, and 13). In addition to the $_3RKR_5$ motif, mammalian p12 possesses yet another multibasic motif $_{15}KKR_{17}$ (Fig 3A, colored in blue). As R3A, K4A, R5A p12 mutant failed to form any oligomer in the presence of formaldehyde, it also implies that $_{15}KKR_{17}$ sequence has no role in dimerization, and the dimeric property of p12 is specifically attributed by the $R_3K_4R_5$ motif. Even the ITC assay failed to detect any binding between wild-type p12 and R3A, K4A, R5A p12 mutant (Fig 1C, ii).

Because we did not detect dimerization of the RKR motif mutant in any of our assays, we wanted to rule out the possibility that this effect resulted from a significant change in p12 conformation. For this reason, we compared the circular dichroism (CD) spectra of the wild-type and R3A, K4A, R5A mutant p12 proteins (Fig 5B). The CD spectra determined in the "far-UV" region (200 to 260 nM) showed p12 to be enriched in α-structure as evident from the characteristic negative peaks at 208 nm and 222 nm. It also indicates that the mutant protein retains a similar level of secondary structures as that of the wild-type protein, which would suggest that the mutations do not cause a major perturbation to the p12 structure.

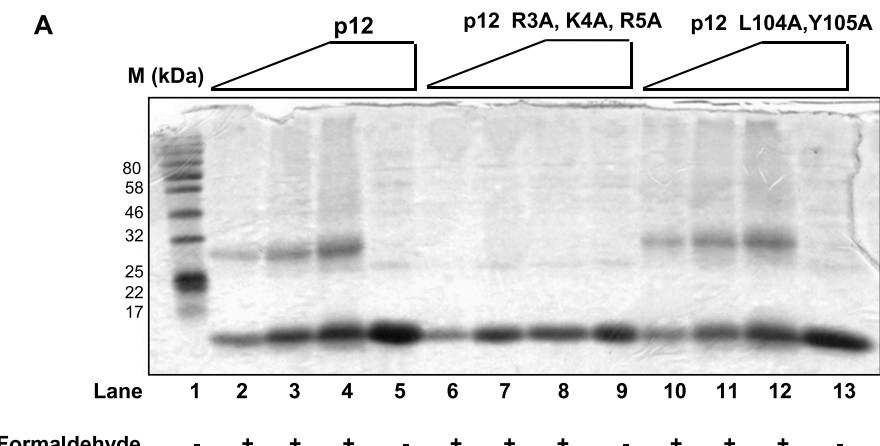

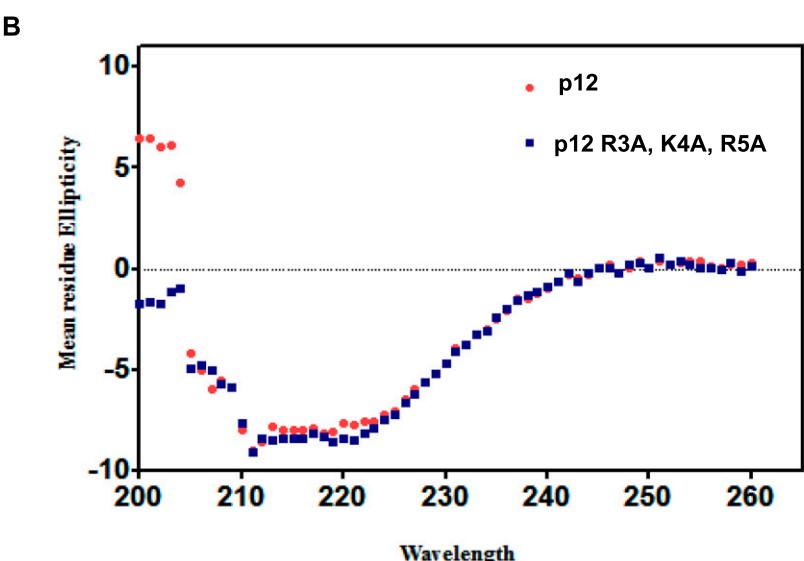

**Figure 5. Formaldehyde cross-linking of p12 proteins. (A)** About 1–5 μg of p12 proteins were cross-linked with 0.5% formaldehyde solution for 30 min at 25°C. After termination with SDS sample buffer, they were resolved in a 12% SDS–PAGE. Lane 1: MW; lanes 2–5: p12; lanes 6–9: R3A, K4A, R5A; and lanes 10–13: L104A, Y105A. Lanes 5, 9, and 13: proteins treated similarly but without formaldehyde. **(B)** Far UV-CD spectra of wild-type (red) and R3A, K4A, R5A mutant (blue) p12. CD spectra at pH 7.5 between 200 and 260 nm were recorded. Data represent values determined after solvent correction and after averaging each set (n = 3).
Source data are available for this figure.

Considering all these pieces of evidence, we conclude that p12 forms a dimer solely mediated by the $R_3K_4R_5$ motif.

### In vitro reconstitution of pentameric human Polδ holoenzyme

Various subassemblies of hPolδ such as p125 alone, p125-p50, p125-p50-p68, and p125-p50-p68-p12 have been purified in vitro by mixing various combinations of purified proteins (Xie et al, 2002; Zhou et al, 2012). In this study, Polδ holoenzyme was expressed by co-transforming two bacterial expression constructs GST-p125 and pCOLA234 (p50-p68-His-FLAG-p12), and the complex was purified using glutathione–sepharose beads to near homogeneity. Taking advantage of the strategically located PreScission protease site, the cleaved Polδ complex was obtained, in which only p12 subunit was amino-terminally FLAG-tagged. We refer this complex as the first Polδ complex. To conclusively show the two different forms of p12 in the holoenzyme, untagged p12 protein purified using bacterial GST-

p12 system was mixed to the first Polδ complex and the mixture was incubated at 4°C for 4 h. If p12 forms a dimer in the Polδ complex, untagged p12 will compete out some of the resident FLAG p12 and a reorganized Polδ with five subunits will appear. Thus, the mixture was loaded into $S_{200}$ mini-column for separation and the fractions were collected in a 96-well plate. As we could not detect enough protein by Coomassie staining despite our repeated trials, various fractions were analyzed in SDS–PAGE followed by detection of various subunits by probing with a specific antibody (Fig 6). The membrane was probed with an anti-p68 antibody (A) and selected fractions again resolved in SDS–PAGE (B) to detect the presence of enriched holoenzyme fractions. As depicted in the figure, initial fractions were enriched in Polδ holoenzyme (A8–B2) and a clear shift of untagged p12 proteins towards the complex was also noticed. Interestingly, early fractions such as A8–A10 showed a majority of FLAG-p12 and very little amount of untagged p12 (first complex), whereas the later fractions convincingly showed the

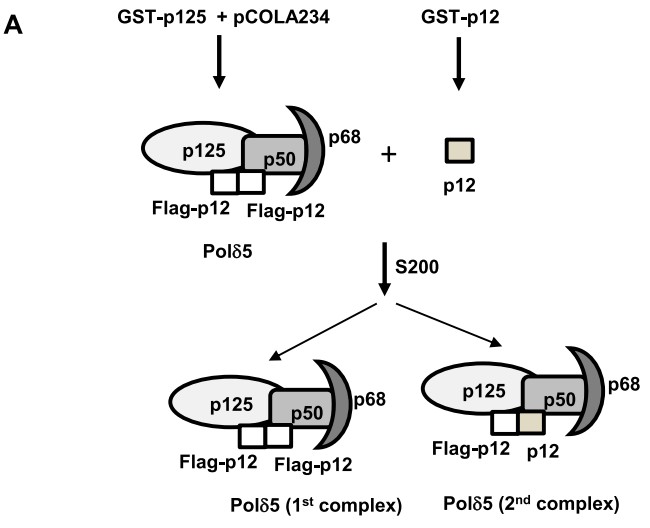

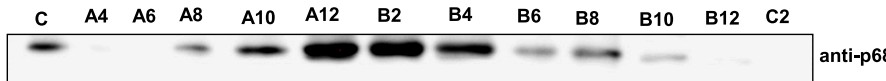

**Figure 6. Purification of the pentameric hPolδ holoenzyme.**
Schematic representation of purification of pentameric Polδ was shown. A mixture of Polδ4 and p12 was separated by gel filtration chromatography, and various fractions were first analyzed by probing with anti-p68 antibody. **(A)** Enriched fractions were again separated in SDS–PAGE and further transferred to PVDF membrane. The membrane was cut into pieces as per the molecular weight of various subunits and was individually probed with a specific antibody. **(B)** The fractions possessing Polδ5 are denoted. Name of the fractions is as per the collections in the 96-well plate.
Source data are available for this figure.

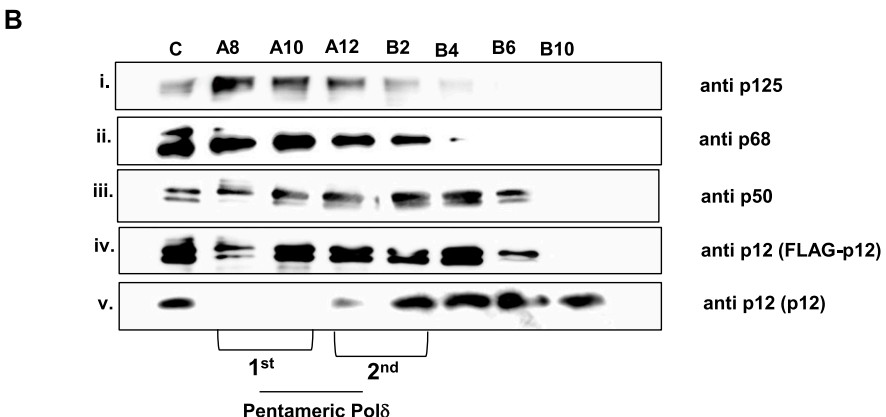

presence of both tagged and untagged p12 subunits (A12–B2, second complex). The faster elution of the first complex from the size-exclusion column suggests that the early eluate is also a pentameric complex. In this approach, we could purify two populations of Polδ5: (a) early Polδ5 fractions containing dimeric FLAG-p12 (A8–A10) and (b) later Polδ5 fractions where some of the FLAG-p12 were replaced with untagged p12 (A12–B2). This analysis demonstrates that Polδ5 is the predominant form and it is unlikely that one can purify Polδ complex with monomeric p12 (Polδ4). Thus, in our opinion earlier reported, purified and functionally characterized Polδ4 holoenzyme could very well be the Polδ5 complex (11, 15, 23, 24, and 39). Considering both our in vivo and in vitro analyses, we propose that hPolδ is intrinsically a pentameric complex, possessing two p12 subunits.

### Dimerization of p12 is essential for PCNA interaction

Because both $R_3K_4R_5$ and $_{98}QCSLWHLY_{105}$ motifs are required for PCNA interaction and the former motif additionally is involved in protein dimerization, to ascertain any regulatory role of this motif in p12 function, a PCNA overlay experiment was carried out. Proteins were resolved on native PAGE to keep the natural folding and the dimer structure intact (as carried out in Fig 4A), transferred to PVDF membrane and blocking was performed in the presence of PCNA. After several washings, the blot was developed with the anti-PCNA antibody (Fig 7A). As depicted in the figure, although p12 and its L104A, Y105A mutant formed dimers (upper panel, compare lanes 1 and 3), a signal for only wild type was detected (lower panel), suggesting that the $_{98}QCSLWHLY_{105}$ motif is the true PIP motif required for PCNA interaction. Despite retaining the PIP motif in the extreme C-terminal tail, because of its inability to form a dimer, the R3A, K4A, R5A mutant failed to bind to PCNA. Similarly, a pull-down experiment was also carried out by taking a mixture of stoichiometry equivalents of purified GST-p12 or various p12 mutants and PCNA in Tris-buffer containing 150 mM NaCl salt concentration (Fig 7B). The mixture was incubated with GST beads at 4°C for 3 h in a rocking condition; beads were washed thrice and the bound PCNA was eluted by SDS-containing sample buffer. The eluted PCNA could be detected by anti-PCNA antibody only when it was mixed with wild-type p12 but not with the L104A, Y105A or R3A, K4A, R5A mutant (compare lane 3 with lanes 6 and 9).

ITC assay was carried out by placing p12 (10 $\mu$M) in the sample cell and titrated by PCNA (120 $\mu$M) injection to determine the binding affinity of p12 with PCNA. We did not observe any significant change in heat when p12 or PCNA was injected to the cell containing a buffer (Fig S4). Upon p12-PCNA binding, the heat was liberated and kinetic parameters such as ΔH, ΔG, and the $K_D$ for the complex were recorded as –80 kcal/mol, –6.80 kcal/mol, and 10 $\mu$M, respectively (Fig 7C). In a similar assay condition, no significant heat exchange was observed in a titration where RKR- or PIP-motif p12 mutants were kept in the cell and PCNA in the syringe, suggesting no detectable interaction between the proteins. Thus, our results suggest that dimerization at the $R_3K_4R_5$ motif promotes p12 interaction with PCNA via the $_{98}QCSLWHLY_{105}$ motif.

### Interdomain connecting loop region of hPCNA mediates its interaction with p12

As our predicted model structure showed p12 PIP peptide binding to IDCL domain of hPCNA, yeast two-hybrid assay was carried out with p12 fused to Gal4-binding domain and two PCNA mutants, namely, *pcna-79* and *pcna-90*, fused to *Gal4* activation domain. In *pcna-79*, two key hydrophobic residues L126 and I128 of the interdomain connecting loop were mutated to alanines, whereas *pcna-90* possesses the two mutations P253A and K254A in the extreme C-terminal tail of PCNA. Most of the interacting proteins bind to any of these two regions of a trimeric PCNA ring (Manohar & Acharya, 2015). While wild type and *pcna-90* were able to interact with p12 as evident from the growth on the SDA plate lacking leucine, uracil, and histidine (Fig 8A, sectors 1 and 3), *pcna-79* did not support the survival as it failed to form intact Gal4 by interacting with p12 (sector 2). The p12 PIP mutant was used as a negative control (sector 4). To strengthen our finding, GST pull-down assay was carried out. An equal stoichiometry of wild type and IDCL mutant of PCNA proteins was incubated with GST-p12 and pull-down assays were performed on glutathione–sepharose affinity beads as described previously (Acharya et al, 2005). As can be seen from the data shown in Fig 8B, although GST-p12 was able to pull down most of the wild-type PCNA from solution (compare lane 1 with lane 3), it failed to bind pcna-79 protein (compare lane 4 with lane 6) as detected by the anti-PCNA antibody.

### Cdm1, a p12 homologue of *S. pombe* also forms a dimer

The other fourth subunit of DNA polymerase δ that has been well characterized is Cdm1, a p12 orthologue from *S. pombe* (Zuo et al, 2000). Cdm1 consists of 160 aa, has a molecular mass of 18.6 kD, and a pI of 7.73. Like p12, it shows abnormal mobility in the SDS–PAGE and migrates as ~22-kD protein (Fig 9A). As it also has a conserved RKR motif ($K_2K_3R_4$ in Cdm1) at its N-terminal end, we wanted to examine whether the homodimerization property of the smallest subunits of Polδ is evolutionarily conserved. Wild-type and K2A, K3A, R4A mutant Cdm1 proteins were purified to near homogeneity from bacterial overexpression system and analyzed on native PAGE. Just like p12, the wild-type Cdm1 migrated slower than its mutant. The slower migration of Cdm1 complex in comparison with p12 dimer could be attributed to the differences in their pI and MW (pI 7.3 versus 6.3; MW 12 versus 18.6). The dimeric p12 incidentally migrates at about the same position with monomeric Cdm1-mutant protein (Fig 9B). Thus, we concluded that homodimerization is the intrinsic property of the fourth subunit of Polδ and is mediated by conserved basic amino acids located at the extreme amino terminal end (RKR/KKR motif).

## Discussion

DNA polymerase δ is a high-fidelity essential DNA polymerase. It not only plays a central role in DNA replication but also participates in DNA recombination and several DNA repair pathways from yeasts to human (Hindges & Hubscher, 1997; Burgers, 1998).

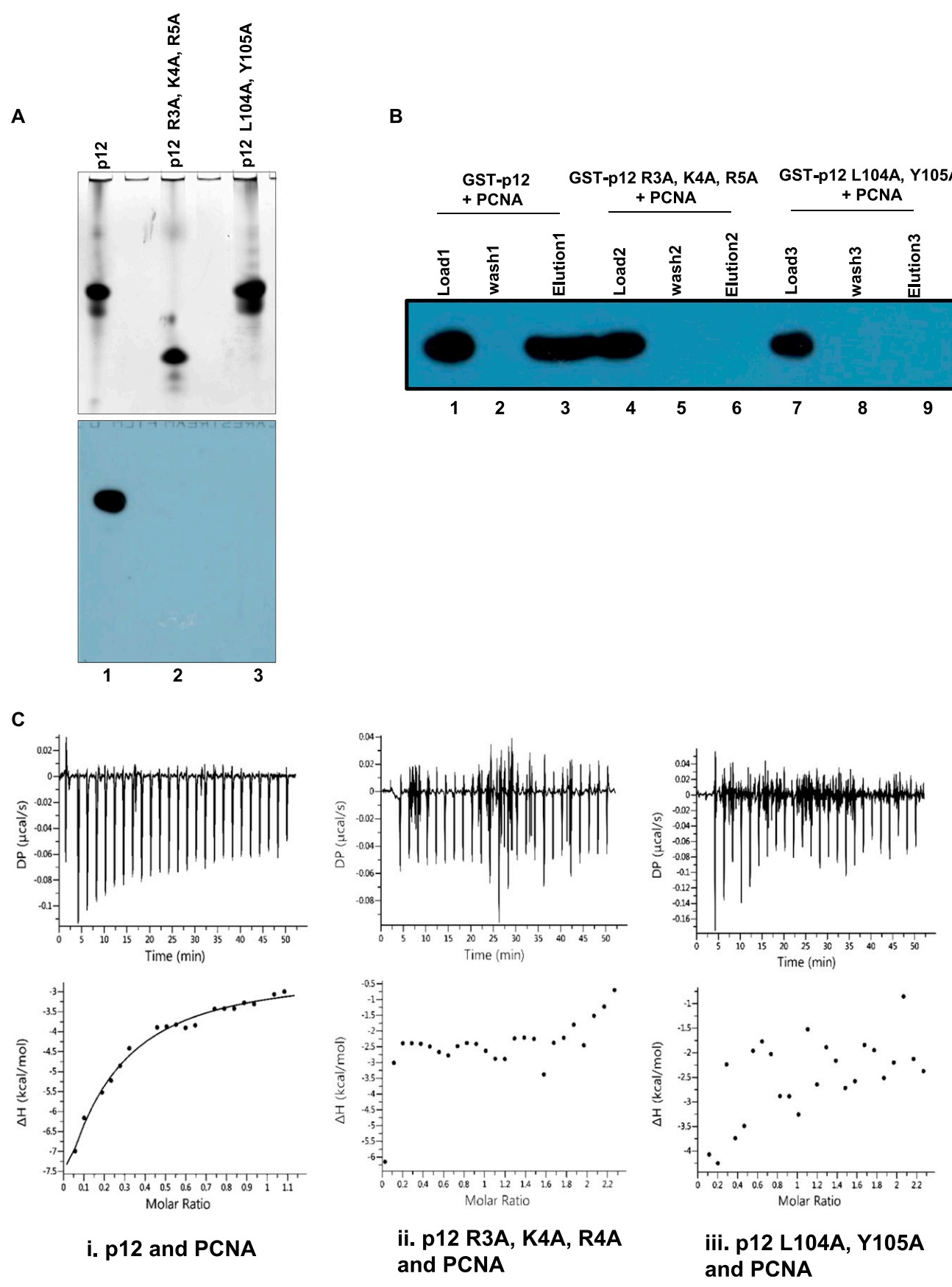

**i. p12 and PCNA**

**ii. p12 R3A, K4A, R4A and PCNA**

**iii. p12 L104A, Y105A and PCNA**

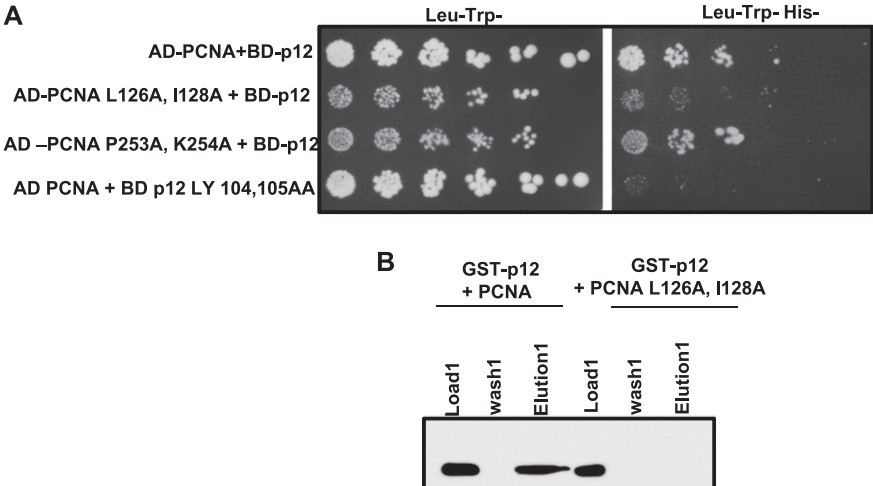

**A**

Leu-Trp-    Leu-Trp- His-

AD-PCNA+BD-p12    1

AD-PCNA L126A, I128A + BD-p12    2

AD –PCNA P253A, K254A + BD-p12    3

AD PCNA + BD p12 LY 104,105AA    4

**B**

GST-p12 + PCNA    GST-p12 + PCNA L126A, I128A

Load1  wash1  Elution1  Load1  wash1  Elution1

1  2  3  4  5  6

**Figure 8.  IDCL of hPCNA binding to p12.**
**(A)** Yeast two-hybrid analysis showing the interaction of PCNA with p12. HFY7C yeast transformants with various GAL4-AD and GAL4-BD fusions were selected on SD media plates lacking leucine and tryptophan, and with and without histidine amino acids. Row 1: AD-PCNA + BD-p12; row 2: AD-PCNA L126A, I128A + BD-p12; row 3: AD-PCNA P253A, L254A + BD-p12; and row 4: AD-PCNA + BD-p12 L104A, Y105A. **(B)** GST pull down of PCNA by wild-type p12. Beads of GST-p12 were mixed with wild-type (lanes 1–3) or L126A, I128A mutant PCNA (lanes 4–6) in equilibration buffer after the incubation beads were washed, and the bound PCNA was eluted by the protein loading dye. Various fractions were resolved in 12% SDS–PAGE, blotted to the membrane, and developed by the anti-PCNA antibody. Lanes 1 and 4: 10% of load; lanes 2 and 5: 10% of third washings; and lanes 3 and 6: total elutes.
Source data are available for this figure.

Several mutations in mouse and hPolδ subunits have been mapped to cause various cancers (Flohr et al, 1999; Goldsby et al, 2002; Albertson et al, 2009). Thus, it is important to understand the function of each subunit and their precise role in processivity and fidelity of the holoenzyme. In this report, we have reinvestigated the role of the smallest subunit of hPolδ, p12, in the holoenzyme architecture and PCNA interaction.

As reported earlier and also from this study, we understand that p12 subunits interact with p125 and p50, whereas p50 makes a connecting bridge between p125 and p68 subunits (Lee et al, 2017). Thus, hPolδ is widely considered to be a heterotetrameric holoenzyme. Our yeast two-hybrid assays, cellular co-localization assay in replication foci, and co-immunoprecipitation assays suggest an oligomerization status of the p12 subunit. Using several physiobiochemical and mutational analyses of p12 proteins, further, we extrapolate our in vivo analyses to suggest that p12 exists as a homodimer in vitro mediated by the $_3RKR_5$ motif, which argues against the tetrameric nature of hPolδ. As it has been previously shown, various subassemblies of hPolδ holoenzymes could exist in the cell based on Polδ's function in either replication or repair (Zhang et al, 2016). We propose that among these sub-complexes, pentameric Polδ is the native form of Polδ. Pull down of cellular Polδ by a tagged p12 and in vitro reconstitution of Polδ5 substantiates our prediction (Figs 2A and 6). This is also supported by the fact that p12 was always present in higher stoichiometry in comparison with other subunits in Polδ preparations by several other groups (Podust et al, 2002; Xie et al, 2002; Wang et al, 2011;

Zhou et al, 2011; Zhang et al, 2016; Lee et al, 2017). We also found that the dimerization of the fourth subunit of Polδ is not restricted to human, as Cdm1 of SpPolδ also forms a dimer, which is again dependent on the KKR motif. As the RKR/KKR motif has been retained in other p12 homologues as well, it appears that such a property of the smallest subunit of Polδ is evolutionarily conserved. Interestingly, the small accessory subunit of yet another B-family polymerase Polζ, Rev7 is found to function as a dimer (Rizzo et al, 2018). Thus, the subunit dimerization of B-family DNA pols could be an intrinsic property, and it could be advantageous for the DNA polymerases to establish multiligand interactions during replication.

Most of the PCNA-interacting partners, including Polδ, bind to PCNA through a structurally conserved canonical PIP motif (Bruning & Shamoo, 2004). Previously, we have shown that all the three subunits of ScPolδ contribute to PCNA interaction as well as its processive DNA synthesis (Acharya et al, 2011). The PIP motifs in Pol3, Pol31, and Pol32 have been mapped, and each of them can bind to a monomer of the trimeric PCNA. However, the scenario of hPolδ interacting to PCNA looks complex, and it is not clear which subunits primarily contribute to the PCNA interaction and the processivity of the enzyme. Although reports suggested that all the four subunits bind to PCNA and contribute enzymatic processivity to different degrees, identification of PIPs remains elusive (Lee et al, 2012). The PIP motif in p125 is yet to be identified. Like the Pol32 subunit, p68 possesses a conventional PIP motif at the extreme carboxyl terminal end (Bruning & Shamoo, 2004), and accordingly,

**Figure 7.  Mutations in RKR motif inhibit binding with PCNA.**
**(A)** The upper panel depicts Coomassie blue–stained gel of various p12 proteins resolved in a non-denaturing condition, whereas the lower panel is a far-Western analysis of a similar gel. Proteins were transferred from the gel to the membrane, and further, the blot was blocked with PCNA. After washings, the bound PCNA was detected by the anti-PCNA antibody. Lane 1: wild-type; lane 2: R3A, K4A, R5A; and lane 3: L104A, Y105A p12 proteins. **(B)** GST pull down of PCNA by wild-type p12. Beads of GST-p12 (lanes 1–3), GST-R3A, K4A, R5A (lanes 4–6), or GST-L104A, Y105A p12 mutants (lanes 7–9) were mixed with PCNA in equilibration buffer after the incubation beads were washed and the bound PCNA was eluted by the protein loading dye. Various fractions were resolved in 12% SDS–PAGE, blotted to the membrane, and developed by the anti-PCNA antibody. Lanes 1, 4, and 7: 10% of load; lanes 2, 5, and 8: 10% of third washings; and lanes 3, 6, and 9: total eluates. **(C)** ITC analysis of binding of wild-type (i), R3A, K4A, R5A (ii), or L104A, Y105A mutant (iii) p12 to PCNA. In each panel, the upper half shows the measured heat exchanges during each PCNA protein injection. The lower half of each panel shows the enthalpic changes as a function of the molar ratio of the two proteins where p12 was considered as a dimer and PCNA as a trimer.
Source data are available for this figure.

**A**

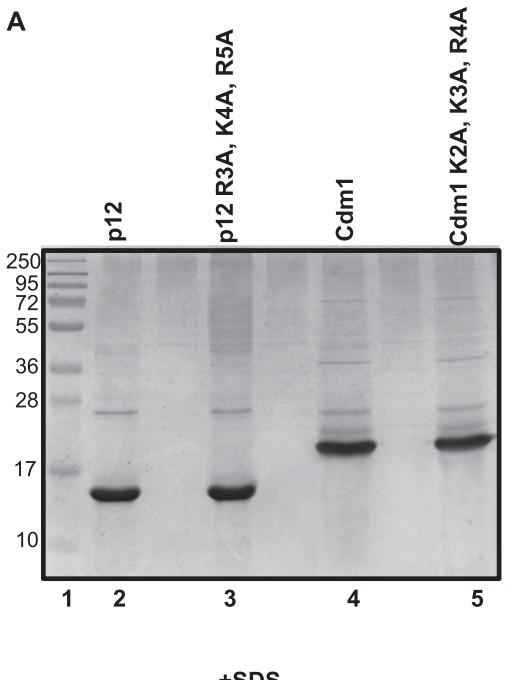

**B**

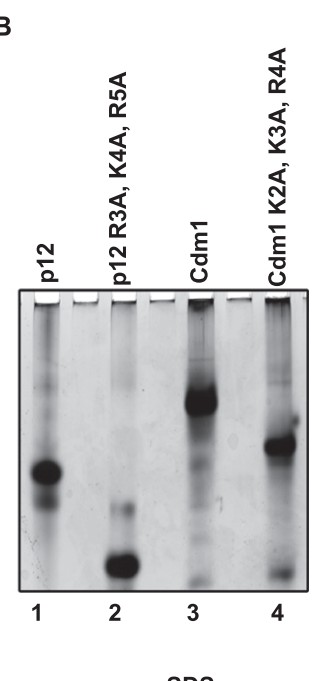

+SDS

-SDS

**Figure 9. Cdm1 also dimerizes via KKR motif and stability of the dimers.**
**(A)** The purity of wild-type and K2A, K3A, R4A mutant Cdm1 proteins were analyzed in 12% SDS–PAGE, and their mobility was compared with p12 proteins. **(B)** These proteins were further resolved in PAGE without SDS. Lane 1: p12; lane 2: p12 with R3A, K4A, R5A mutations; lane 3: Cdm1; and lane 4: Cdm1 with K2A, K3A, R4A mutations. Source data are available for this figure.

deletion of last 20 aa of p68 that encompass PIP sequences failed to bind to PCNA. Another study also revealed a mechanism that could modulate the interaction of p68 with PCNA by a protein kinase–mediated phosphorylation of Ser458 in the PIP-box $_{456}$QVSITGFF$_{463}$ (Lemmens et al, 2008). Similarly, a 22-aa oligopeptide containing the PIP sequence $_{57}$LIQMRPFL$_{64}$ of p50 was shown to bind PCNA by far-Western analysis (Lu et al, 2002; Wang et al, 2011); however, mutational analysis in this motif to provide functional evidence is yet to be carried out. Extensive biochemical studies carried out in the past suggest that the absence of p12 impedes processive DNA synthesis of Polδ. Here, by using structural modeling, yeast two-hybrid system, and many other biophysical and chemical studies, we have mapped the PIP motif of p12 to be at the C-terminal tail that forms a typical 3$_{10}$ helix and interacts with IDCL of PCNA. In addition, we showed that whereas the C-terminal sequence $_{98}$QCSLWHLY$_{105}$ is a PIP motif that is directly involved in PCNA interaction, the N-terminal motif $_4$KRLITDSY$_{11}$ is involved in dimerization and that it indirectly participates in p12 interaction with PCNA. Contrary to an earlier observation, our study reveals that the role of the RKR motif in PCNA interaction is mostly indirect as the monomeric form of p12 does not bind to PCNA (Li et al, 2006).

There are sufficient pieces of evidence, both in vitro and in vivo, to support the idea that multiple subassemblies of Polδ may exist. Proteolysis of p12 and p68 subunits by human calpain-1 could trigger interconversion of Polδ in the cell (Rahmeh et al, 2012; Terai et al, 2013; Zhou et al, 2012). Based on the dimerization of p12 and the purification of Polδ5, we propose existence of four different complexes of Polδ: Polδ5 (p125+p50+p68+2xp12), Polδ3 (p125+p50+p68), Polδ2 (p125+p50), and p125 alone (Fig 10A). Depending upon the cellular contexts such as either actively replicating cells or cells under genomic stress, this interconversion between the Polδs

might happen. Polδ5 could be the major holoenzyme that takes part in DNA replication, which was earlier extensively studied as Polδ4. Our in vitro reconstitution assay rules out the purification of Polδ4 with a monomeric p12; thus, it may not exist in the cell. So, now instead of four subunits, five subunits of Polδ should be considered that will interact with the three available IDCLs in the trimeric PCNA, unless they use other binding sites such as inter-subunit junction or the C-terminal domain of PCNA (Eissenberg et al, 1997; Gomes & Burgers, 2000). Genetic analyses of Polδ PIPs in *S. cerevisiae* revealed that for cell survival, along with Pol32 PIP, any one among the Pol3 or Pol31 PIPs is essential. In the absence of functional Pol32 PIP domain, PIP domain mutation in Pol3 or Pol31 subunits causes lethality (Acharya et al, 2011). Despite being structural subunits, *CDC27* and p68, the Pol32 homologues are essential in *S. pombe* and mice, respectively (Murga et al, 2016), and it explains their major role in Polδ function in DNA synthesis. The binding affinity of Polδ with PCNA also increases when p12 or p68 binds to the core (Kd = 8.7~9.3 nM), and it further increases when all the subunits are present together (Kd = 7.1 nM). The Kd of Polδ core is found to be 73 nM (Lee et al, 2017). Accordingly, the addition of p68/Pol32 to the core (p125+p50 or Pol3+Pol31) results in high processivity; thus, its binding to PCNA appears to be critical. Considering all these, we propose a modified model for the network of protein–protein interactions of the Polδ–PCNA complex (Fig 10B). In a pentameric state of hPolδ, along with p68, any other two subunits among p125/p50/p12 will bind to PCNA in any combinations as shown in Fig 10B (i–iv). Upon p68 degradation or its phosphorylation, p125, p50, and one monomer of p12 dimer can bind to PCNA (v). Similarly, upon p12 proteosomal degradation as a response to DNA damage; the other three subunits will make contacts with PCNA (vi), whereas because of cleavage of both p68 and p12 in certain situations, p125 and p50

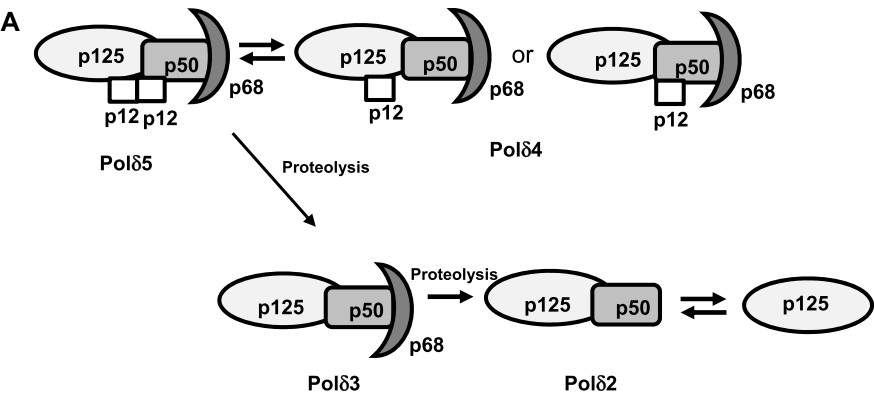

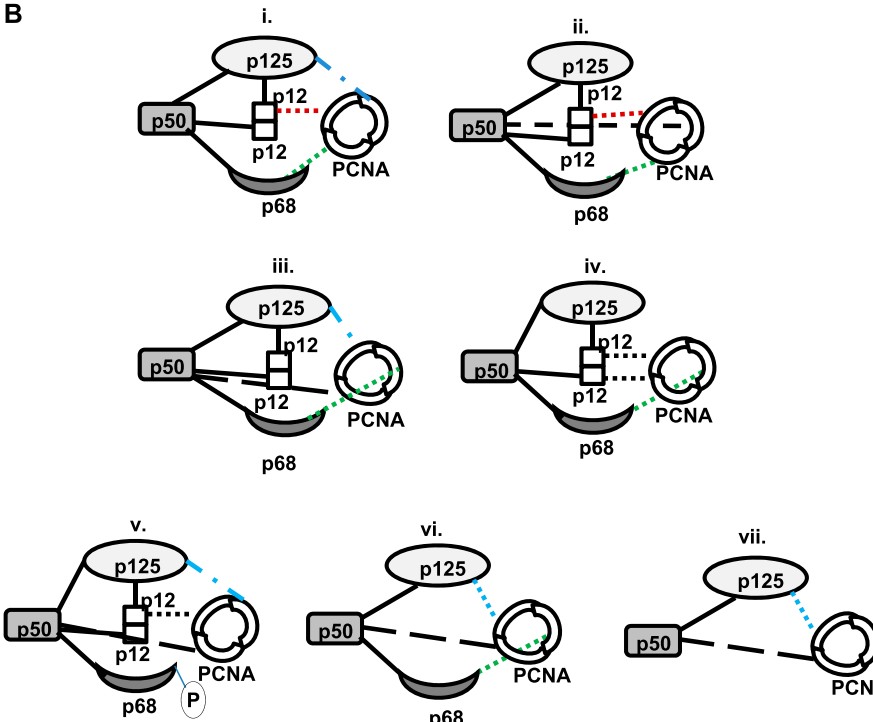

**Figure 10. Protein–protein network model for hPolδ and PCNA.**
**(A)** Depicting different subassemblies of Polδ complexes. As p12 is a dimer, pentameric Polδ holoenzyme is proposed to function in DNA replication. Polδ4 may or may not exist in the cell; however, after proteolysis during certain conditions such as genomic stress, Polδ5 can be downgraded to Polδ3 or Polδ2 complexes. **(B)** Four different proposed modes of Polδ5 binding to PCNA, where apart from p68, any other two subunits can bind to IDCLs of PCNA. In the dimerization state, only p12 can bind to PCNA (i–iv). Upon phosphorylation of p68 or proteolysis of p12, other remaining three subunits bind to PCNA (v and vi). In the case of the core, both subunits interact with PCNA but with compromised processive DNA synthesis. PCNA monomers binding to p125, p50, p68, and p12 are shown in blue, black, red, and green dotted lines, respectively. Source data are available for this figure.

bind to PCNA (vii). Thus, this study warrants extensive mutational analyses as was carried out in ScPolδ, rather than analyzing various sub-complexes to decipher the precise role of these PIPs in cellular function and processive DNA synthesis by hPolδ.

In conclusion, here we show that RKR-mediated dimerization plays a vital role in p12 binding to PCNA and Polδ5 architecture, and the phenomenon appears to be conserved throughout evolution.

## Materials and Methods

### Plasmids, oligonucleotides, antibodies, and enzymes

Human DNA polymerase δ constructs pET32-p125 and pCOLA-hPold234 (a kind gift from Prof. Y Matsumoto) were used as precursor plasmids for subsequent manipulation (Schmitt et al, 2009). The oligonucleotides (Table S1) from Integrated DNA Technologies (IDT), Q5 high-fidelity DNA polymerase and other restriction enzymes from NEB, and antibodies from Sigma or Abcam were procured. Human PCNA was directly amplified from cDNA synthesized from total RNA of HELA cell line by using primers NAP239 and NAP240 and cloned into a pUC19 vector. The PCR product was digested with BamHI and cloned into the BglII site to generate a bacterial expression system of GST-fused hPCNA. To express in budding yeast, hPCNA was amplified by using primers NAP251 and NAP240, the PCR product was digested with BamHI and cloned into the same site of pGBT9 and pGAD424 to generate Gal4BD-hPCNA and Gal4AD-hPCNA, respectively. Inverse PCR was carried out using the primer set NAP300-NAP304 on pUC19-hPCNA to generate pcna-79 (L126, 128AA), and further subcloned into an expression vector. Human pcna-90 (P253A, K254A) was PCR-amplified using primers

NAP251 and NAP305 from the pUC19-hPCNA template; the PCR product was digested with BamHI and cloned into the same site of pUC19 and pGBT9. A GFP-fused hPCNA expression construct under the CMV promoter was a gift from Prof. Wim Vermeulen.

Amplified PCR products of wild type; R3A, K4A, R5A; and L104A, Y105A mutants of p12 from pCOLA-hPold234 by using primer pairs NAP261-NAP262, NAP362-NAP261, and NAP265-NAP262, respectively, were digested with EcoRI–BamHI and cloned into the same sites of pGBT9 and pGAD424. Similar PCR products amplified using the primers NAP260-NAP261, NAP373-NAP261, and NAP265-NAP260 were also digested with BamHI and cloned into the BglII site of either pNA716 or p3X FLAG CMV7.1 vector to generate N-terminal GST-tag proteins expressed under T7 promoter and the N-terminal FLAG-tag human cell expression system, respectively. For confocal micros-copy and pull-down studies, BamHI fragments of various p12 were subcloned into the BamHI site of the pcDNA-GFP vector. For RFP-fusion, the p12 fragment and the catalytic domain of Polθ were amplified with the primer sets NAP448-NAP450 and NAP444-NAP151, respectively, digested with EcoRI–BamHI, and cloned into the same sites of the pASred2-c1 vector.

Other hPolδ subunits such as p125, p50, and p68 were also PCR-amplified by the primer sets NAP252-NAP248, NAP254-NAP255, and NAP258-NAP257, respectively, digested with EcoRI–BamHI, and cloned into the same sites of pGBT9. Wild-type and K2A, K3A, R4A mutant of Cdm1 were PCR-amplified from *S. pombe* genomic DNA using primers NAP361-NAP451 and NAP361-NAP452, respectively, and the BamHI-digested products were cloned into the BglII site in pNA716 for bacterial expression. Similarly, GST-hp125 expression plasmid was generated by cloning a BamHI-digested PCR product amplified from pET32-p125 as a template using NAP247 and NAP248 primers into the BglII site of pNA716. All these constructs were authenticated by DNA sequencing.

## GST-fusion protein purification

All GST-tagged proteins were expressed in *Escherichia coli* BL21 DE3 cells and purified by affinity chromatography using glutathione–sepharose beads (GE Healthcare). The proteins were expressed as amino-terminal GST-fusion proteins under a T7 promoter. Briefly, 5-ml pre-culture of the transformant was added to 500 ml LB + 50 $\mu$g/ml ampicillin and grown at 37°C till the $OD_{600}$ reaches 0.6. Next, the culture was induced with 1 mM IPTG and allowed to grow for an-other 8 h. Cells were harvested, and about 3 gm of frozen cells were resuspended in 1× cell breaking buffer (50 mM Tris–HCl, pH 7.5, 10% sucrose, 1 mM EDTA, 500 mM NaCl, 0.5 mM PMSF, 0.5 mM benza-midine hydrochloride, 10 mM $\beta$-mercaptoethanol, and protease inhibitor cocktail). The cells were lysed with a high-pressure ho-mogenizer at 10000 psi (Stansted). The lysate was cleared by centrifugation at 10000 rpm for 10 min. Furthermore, the super-natant was centrifuged at 30000 rpm for 1 h in a P70AT rotor (Hitachi). Rest of the steps used for purification was the same as described before (Acharya et al, 2005). All the proteins were stored in the buffer containing a final concentration of 50 mM Tris–HCl (pH 7.5), 150 mM NaCl, 10% glycerol, 5 mM DTT, and 0.01% NP-40. The purity of the protein was confirmed after resolving on 12% SDS–PAGE and stained by Coomassie blue.

However, for hPolδ purification, bacterial strain BLR (DE3) was co-transformed with pLacRARE2 plasmid from Rosetta2 strain (Novagen), GST-p125, and pCOLA-hPold234; and colonies were se-lected on LB agar plates containing ampicillin (50 $\mu$g/ml), kana-mycin (30 $\mu$g/ml), and chloramphenicol (35 $\mu$g/ml). About 60 ml overnight-grown pre-culture was inoculated into 6 liters LB with mentioned antibiotics and grown at 37°C to an $OD_{600}$ of 0.6, fol-lowed by induction with 1 mM IPTG, and further growth was con-tinued for 15 h at 16°C. The cells were harvested and stored at –80°C until use. The cell breaking condition and other purification steps were followed as mentioned above. Taking advantage of the strategically located PreScission protease site, cleaved Polδ4 (p125-p50-p68-p12) was obtained in which only p12 subunit was amino-terminally FLAG-tagged. Similarly, p12 protein was also purified by using bacterial GST-p12 construct.

## Size-exclusion chromatography

For size-exclusion chromatography, about 10 $\mu$g of each freshly purified p12 protein was loaded onto a Superdex 200 PC3.2/30 mini-column pre-equilibrated with a buffer containing 50 mM Hepes (pH 7.5), 150 mM NaCl, and 10% glycerol. Chromatography was per-formed twice on an AKTA pure M system (GE Healthcare) at a flow rate of 0.05 ml/min at 4°C, and the absorbance was monitored at 280 nm.

## Purification of human Polδ5 complex

To purify Polδ5 complex; pre-purified Polδ4 (10 $\mu$g) was mixed with an equal amount of untagged p12 protein under rocking condi-tions for 4 h at 4°C. Then, the mixture was injected into a Superdex 200 PC3.2/30 minicolumn pre-equilibrated with a buffer con-taining 50 mM Hepes (pH 7.5), 150 mM NaCl, and 10% glycerol. Chromatography was performed on an AKTA pure M system (GE Healthcare) at a flow rate of 0.03 ml/min at 4°C, and the ab-sorbance was monitored at 280 nm. The eluate was collected in a 96-well plate fraction collector and then the various fractions were subjected to SDS–PAGE and Western blot analysis for the detection of individual subunits of hPolδ. The experiment was repeated thrice for Coomassie blue staining, but we could not detect any protein in the fractions. However, our Western analysis consistently reproduced the same results. To check the presence of Polδ fractionation, the membrane was first probed with an anti-p68 antibody (Cat. No. WH0010714M1) because it does not directly interact with p12 and will give a clear indication of the complex. Furthermore, the enriched fractions were analyzed by probing with a specific antibody such as anti-p125 (Cat. No. SAB4200053), anti-p50 (Cat. No. SAB4200054), and anti-p12 (Cat. No. WH0057804M1). Horseradish peroxidase–conjugated host-specific secondary IgG (Cat. No. A90376154; Sigma–Aldrich and Cat. No. W402B; Abcam) was used to develop the blot by Chemidoc.

## Yeast two-hybrid analyses

The yeast two-hybrid analyses were performed using *HIS3* as a reporter system (Acharya et al, 2005). The HFY7C yeast strain (from Clonetech) was transformed with various combinations of the

GAL4-AD (TRP1) and GAL4-BD (LEU2) fusion constructs. Co-transformants were obtained on synthetic dropout (SD) media plates lacking leucine and tryptophan. To verify the interaction, the transformants were grown on 5 ml YPD liquid medium overnight at 30°C and various dilutions were either streaked or spotted on Leu⁻Trp⁻His⁻ selection medium. Furthermore, the plates were incubated for 2 d at 30°C and photographed. Yeast transformants exhibiting histidine prototrophy are indicative of protein–protein interaction. The auxotrophic reporter assay was carried out by taking three different co-transformants.

## Formaldehyde cross-linking

About 1–5 $\mu$g of native or mutant p12 protein in 20 mM Hepes buffer (pH 7.5) was mixed with 0.5% formaldehyde solution for 30 min at 25°C. The reaction was terminated by the addition of SDS sample buffer. Cross-linked proteins were resolved by electrophoresis in a 12% SDS–PAGE. The gel was stained with Coomassie blue. The experiment was repeated again with a different batch of purified proteins.

## Native PAGE analysis

Wild-type and various mutants of p12 and Cdm1 proteins were mixed with a DNA loading dye and were analyzed on 12% native PAGE. The gel was run at 80 V for 5 h at 4°C using a running buffer containing 25 mM Tris base and 192 mM glycine, at pH 8.8. The proteins were visualized by Coomassie blue staining of the gels.

## Confocal microscopy

CHO cells were grown up to 70% confluency on the cover glass in DMEM supplemented with 10% FBS and 1× penicillin–streptomycin antibiotics solution. Furthermore, the cells were washed with DPBS (pH 7.4) and then replaced with DMEM containing 5% FBS. These cells were co-transfected with GFP/RFP fusion constructs of p12, PCNA, and a catalytic domain of Pol$\theta$ in various combinations as required by using the Lipofectamine Transfection kit as per the manufacturer's protocol (Invitrogen). Furthermore, the cells were incubated at 37°C with 5% $CO_2$ and 95% relative humidity. After 48 h, the cells were thoroughly washed thrice with DPBS, fixed with ice-cold 100% methanol at –20°C for 20 min, followed by rinsing with DPBS (pH 7.4), and then the slides were prepared using antifade as a mounting agent. Images were taken using Leica TCS SP5 at 63× objective. Three independent experiments were carried out for each co-transfectant.

## Co-immunoprecipitation

HEK293 cells were grown up to 70% confluency in a 10-cm dish containing DMEM supplemented with 10% FBS and 1× penicillin–streptomycin antibiotics. These cells were co-transfected with FLAG-p12 with either GFP-p12 or GFP-p12 R3A, K4A, R5A mutant by using the Lipofectamine Transfection kit. The cells were grown in a humidified $CO_2$ incubator at 37°C. After 48 h of growth, the cells were

harvested, washed thrice with DPBS, and immediately resuspended in RIPA buffer (50 mM Tris–HCl, pH 8.0, 0.5% sodium deoxycholate, 1,000 mM NaCl, 0.1% SDS, 1 mM EDTA, 1 mM EGTA, 25 mM sodium pyrophosphate, 1 mM $\beta$-glycerophosphate, 1 mM sodium ortho-vanadate, and protease inhibitor tablet) and kept for 1 h at 4°C rocking condition. After centrifugation at 10,000 rpm, the supernatant was collected and protein concentration was determined using the Bradford method. About 500 $\mu$g of total protein was incubated overnight with anti-FLAG antibody-conjugated agarose beads. The beads were washed thrice with RIPA buffer, and bound proteins were eluted by 40 $\mu$l of SDS loading buffer and subjected to 12% SDS–PAGE. The proteins from the gel were transferred to PVDF membrane, followed by incubation of the membrane with 5% skim milk in PBST for 1 h at room temperature. The blot was washed thrice with PBST and incubated with the anti-GFP antibody (1:5,000 dilution, Cat. No. ab290; Abcam) for 2 h at RT. Subsequently, after thorough washings, horseradish peroxidase–conjugated goat anti-rabbit IgG (diluted 1:10,000 in PBST, Cat. No. A6154; Sigma-Aldrich) was used to develop the blot.

Similarly, native hPol$\delta$ was co-immunoprecipitated from HEK293 cells transfected with GFP-p12. About 500 $\mu$g of total protein was incubated overnight with anti-GFP or anti-p125 antibody–conjugated agarose beads. The beads were washed thrice with RIPA buffer, and bound proteins were eluted by 40 $\mu$l of SDS loading buffer and subjected to 12% SDS–PAGE. The proteins from the gel were transferred to PVDF membrane, cut into four pieces as per the molecular weight markers, and the membranes were individually incubated with 5% skim milk in PBST for 1 h at room temperature. The blot was washed thrice with PBST and probed with subunit-specific antibody for 2 h at room temperature. For p50 probing, first the membrane was probed with the anti-p50 antibody, then stripped off, and again probed with anti-GFP antibody as both the proteins migrate close to each other. Subsequently, after thorough washings, horseradish peroxidase–conjugated goat anti-IgG (diluted 1:10,000 in PBST, Cat. No. A6154; Sigma-Aldrich) was used to develop the blot.

## PCNA overlay assay

Various proteins were resolved in two 12% native PAGE, and whereas one of the gel developed with Coomassie blue, the other one was transferred to methanol-activated PVDF membrane. The blot was first washed with BLOTTO (25 mM Tris–HCl, pH 7.4, 150 mM NaCl, 5 mM KCl, 5% fat-free milk, 1% BSA, and 0.05% Tween 20) for 1 h at room temperature. Then, the blot was incubated overnight at 4°C in 10 $\mu$g/ml of PCNA containing BLOTTO with constant agitation. After three rinses with BLOTTO, the membrane was incubated with the anti-PCNA antibody (diluted 1:1,000, Cat. No. SAB2108448; Sigma-Aldrich) in BLOTTO. Subsequently, after thorough washings, horseradish peroxidase–conjugated goat anti-rabbit IgG (diluted 1:10,000 in PBST, Cat. No. A6154; Sigma-Aldrich) was used to develop the blot.

## GST pull-down assay

GST wild-type or LY 104,105 AA p12 protein–bound glutathione–sepharose beads were mixed with 0.5 $\mu$g of either wild-type or

mutant (L126A, I128A) human PCNA, and a pull-down experiment was carried out using a standardized protocol described previously (Acharya et al, 2005). Then the beads were thoroughly washed three times with 10 volumes of equilibration buffer (50 mM Tris–HCl, pH 7.5, 150 mM NaCl, 5 mM dithiothreitol, 0.01% NP-40, and 10% glycerol). Finally, the bound proteins were eluted with 50 $\mu$l SDS loading buffer. Various fractions were resolved on a 12% SDS–PAGE, followed by Western blot analysis similarly performed as in co-immunoprecipitation experiment except that the primary antibody used is an anti-PCNA antibody (Cat. No. SAB2108448; Sigma-Aldrich) in 1:750 dilutions.

### Isothermal titration calorimetry

The purified p12 and PCNA proteins were dialyzed overnight in 1 liter of a buffer containing 20 mM Hepes (pH 7.4) and 150 mM NaCl at 4°C to ensure complete removal of DTT and glycerol from the protein storage buffer, which could affect the heat exchange. ITC assays were performed using a Malvern MicroCal PEAQ-ITC calorimeter. Before the experiment, the cell and the syringe were thoroughly washed with water, followed by cell rinsing with a buffer. A control run was carried out to make sure that the buffer is not participating in heat change where the cell was filled with 300 $\mu$l of a buffer and concentrated p12 or PCNA protein (120 $\mu$M) in the syringe. The titration did not show any false binding. ITC was performed using p12 (10 $\mu$M) in the sample cell and PCNA or p12 (120 $\mu$M) in the syringe. Twenty to twenty-five times 1.5–2 $\mu$l of protein from the syringe was injected at intervals of 120 s with an initial delay of 120 s at 25°C. For p12–p12 interaction, the reaction was carried out at 30°C. Because of the binding of the ligand to protein in the cell, in these studies, the heat was generated and the difference of heat changes with respect to the reference cell that only contains water was detected and measured. The data were analyzed to determine various kinetic parameters using a single-site binding model provided in the ITC analysis software package. The experiments were repeated thrice with different batches of purified proteins.

### Circular dichroism

The purified p12 proteins were dialyzed overnight into 1 liter of a buffer containing 20 mM Hepes buffer (pH 7.5) and 20 mM NaCl at 4°C. The secondary structure of p12 and p12 R3A, K4A, R5A mutant was determined by CD spectroscopy using a Chirascan (Applied Photophysics). Spectra were taken at 25°C in a 10-mm path-length quartz cuvette containing the sample at concentrations of 0.2 mg/ml of protein in 20 mM Hepes buffer (pH 7.5) and 20 mM NaCl. The spectra were corrected for the buffer. Mean residue ellipticity values were calculated using the expression $[\theta] = \theta \times 100/(cln)$, where $\theta$ is the ellipticity (in millidegrees), c is the protein concentration (in mol/liter), l is the path length (in centimeter), and n is the number of amino acid residues. The analysis was repeated thrice with different batches of purified proteins.

### In silico analysis of p12 structures

p12 RKR (1-MGRKRLITDSYPVK-14) and PIP (92-GDPRFQCSLWHLYPL-106) domains were used for peptide structure prediction by using

PEP-FOLD3 server (http://bioserv.rpbs.univ-paris-diderot.fr/services/PEP-FOLD3/). The models generated were validated by the SAVES and Ramachandran plot. Furthermore, the generated structural models were aligned with PIP peptide sequences from p21 (1AXC) and p68 PIP (1U76).

## Supplementary Information

## Acknowledgements

We are grateful to Profs. Y Matsumoto and W Vermeulen for providing us hPol$\delta$ expression and CMV-GFP-hPCNA plasmids, respectively. We thank Sitendra Prasad Panda for his technical assistance, Dr. Jawed Alam for his involvement in initial studies, and Bhabani Shankar Sahoo for his help in confocal microscopy. Our laboratory colleagues are acknowledged for their thoughtful discussion. P Khandagale is a DBT senior research fellow; K Manohar and D Peroumal are thankful to CSIR-SRF and DBT-RA fellowships, respectively. This work was supported by the intramural core grant from Institute of Life Sciences,, Bhubaneswar, India.

## Author Contributions

P Khandagale: data curation, resources, formal analysis, validation, investigation, methodology, and writing—review and editing.
D Peroumal: formal analysis, investigation, methodology, and writing—review and editing.
K Manohar: formal analysis, investigation, methodology, and writing—review and editing.
N Acharya: conceptualization, formal analysis, supervision, funding acquisition, validation, visualization, project administration, and writing—review and editing.

### Conflict of Interest Statement

The authors declare that they have no conflict of interest.

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
