## [Reviewer comments · Life Science Alliance]

Life Science Alliance

Human DNA polymerase delta is a pentameric holoenzyme with a dimeric p12 subunit

Prashant Khandagale, Doureradjou Peroumal, Kodavati Manohar, and Narottam Acharya
DOI: <https://doi.org/10.26508/lisa.201900323>

Corresponding author(s): Narottam Acharya, Institute of Life Sciences

Review Timeline:

Submission Date:	2019-01-29
Editorial Decision:	2019-02-20
Revision Received:	2019-02-26
Editorial Decision:	2019-02-27
Revision Received:	2019-03-04
Accepted:	2019-03-11

Scientific Editor: Andrea Leibfried

Transaction Report:

February 20, 2019

Re: Life Science Alliance manuscript #LSA-2019-00323-T

Dr. Narottam Acharya
Institute of Life Sciences
Infectious Disease Biology
Nalco Square
Bhubaneswar, Odisha 751023
India

Dear Dr. Acharya,

Thank you for submitting your manuscript entitled "Human DNA polymerase delta is a pentameric holoenzyme complex with dimeric p12 subunit" to Life Science Alliance. The manuscript was assessed by expert reviewers, whose comments are appended to this letter.

As you will see, the reviewers appreciate your data and they provide constructive input on how to further strengthen it. We would thus like to invite you to provide a revised version of your work, addressing the comments made by all reviewers. Importantly, *in vivo* p12 dimerization and the formation of a pentamer are currently not supported by the data provided.

Thank you for this interesting contribution to Life Science Alliance. We are looking forward to receiving your revised manuscript.

Sincerely,

B. MANUSCRIPT ORGANIZATION AND FORMATTING:

Reviewer #1 (Comments to the Authors (Required)):

The manuscript shows a range of experiments that, taken together, provide convincing evidence that p12 subunit of the Pol Delta holoenzyme is dimeric in solution and when integrated into the holoenzyme. This is an important point that the field should be aware of. Interestingly, they provide evidence that the dimer interaction is required for the ability of p12 to interact with PCNA and map

for the first time the PCNA interacting region. The data and information in the manuscript is definitely publishable and advances our understanding of the Pol Delta complex. There are a number of issues that the authors need to clarify before publication and the manuscript should be shortened.

Page 1. first para. The authors should clarify that pol Epsilon performs leading strand synthesis and Pol delta performs lagging strand synthesis. This has been shown in both yeasts using physical assays. Pol Delta is capable of performing leading strand synthesis when pol Epsilon is absent, and does so in SV40 replication because the CMG helicase, which anchors pol epsilon to the replisome, is not the replicative helicase in these circumstances.

General: The authors discuss two yeasts. In some places they say "in yeast". It needs to be specified which yeast.

The language could do with a further edit. Many figures could be combined to generate a 4 or 5 figure manuscript.

Page 6 para 2 (relating to figure 2). The immunofluorescence data do not distinguish if the GFP and RFP p12 tagged complexes are in the same pol delta complex or different pol delta complexes that are co-located in the same replication factory. These data are therefore over-interpreted in the manuscript. The authors cannot say these data suggest that both proteins are part of a replication unit and function together as an oligomeric protein.

Page 7 para 1. Figure 3C is referred to as figure 4C

Page 10 first line of para 2. involved should be changed to: required for

Figure 1 (also Fig 9 B)

B. Was reducing agent used (if not, justify, i.e. are there any cysteines in p12 (or Crm1))

Figure 2

A. Add sizes for all the bands shown

C. It is unclear what is being measured. Usually ITC injects concentrated protein into a dilute solution. The p12 in the syringe is presumably concentrated (this is not discussed sufficiently in the materials and methods, where more detail for this particular experiment should be given). Thus largely dimeric protein is being injected?. What is producing the heat change? I would have expected to see a plot of the "heat of dilution". i.e. p12 injected into empty buffer. At a minimum this must be discussed as a caveat.

Fig3. A it would be helpful to put the mammalian only consensus above the alignment.

Figure 7C. Again, heat of dilution?

Reviewer #2 (Comments to the Authors (Required)):

The manuscript describes an impressive set of biochemical, biophysical and in silico experiments documenting that human and fission yeast DNA polymerase delta (pold) is a pentamer with a dimeric smallest subunit, p12. The authors mapped amino acids in N-terminus responsible for dimerization and described how alteration of the motif affects the interaction of p12 with PCNA via PIP motif at C-terminus. The observations are new and add to our knowledge of the critical enzyme replicating the human genome. The weakness of the work is the complete absence of functional analyses (i.e., DNA polymerase activity and processivity) of pold with the variants of p12. The discussion could be more concise. We also noticed that the manuscript needs careful editing to avoid typos, jargon (e.g., strains or cell lines are transformed, not plasmids "are transformed") or clumsy wording.

Detailed comments are below.

Abstract.

The equivocal note about "all three DNA transactions" could be better substituted by the concrete "replication, repair and recombination."

The sentence starting from "Our mutational..." contains unexplained abbreviations and thus could be understood only by narrow specialists in the field.

Typo "dimerzes".

Introduction.

Page 3.

In the first sentence, it is better to write "for lowering of the rate of spontaneous mutations and suppressing carcinogenesis."

In the second sentence, we suggest adding references to recent reviews PMID:28301743 and PMID:28749073.

In the sentence that starts with "Pold not only can..." we suggest to consider addition of the reference to the recent relevant study PMID:30451148.

SV40 replication system exploits two pols, not "suggests requirement."

For the sentence starting from "Irrespective..." a more focused reference could be found, e.g., PMID:19296856.

The last sentence of the first paragraph is unclear. Also, some subunits share a partial similarity, for example, dpb3 and dpb4 possess a histone fold.

The first sentence of the second paragraph might be a stretch.

Can the authors rephrase "the deletion is not achievable?"

What is a "genotoxic burden"?

Page 4.

Proofreading, not "proof reading."

The statement in the sentence starting with "Except in p68 and p50..." does not take into account the paper PMID:22119860.

"Therefore" in the last paragraph could be omitted.

Results.

Page 4.

The second sentence should be edited.

The third should be edited too. It is incorrect to say that a subunit "was more than a 1:1 ratio."

Can the authors comment of a relatively high abundance of p68?

Page 5.

Why "co-transformed" in the third sentence?

Did the authors try a second marker (lacZ) for the two-hybrid system?

The last sentence of the top paragraph is not clear.

"Closed" or "close" in the third paragraph?

How one can "decipher" oligomerization?

What is "irrespective of any antibody?"

Page 6.

Second sentence - more explanations of Fig 2A is desirable, the results look similar for both pulldowns.

Second paragraph. How p68 and 50 were "deciphered?"

What is the rationale to use pol theta as a control?

Page 7.

The meaning of the sentence starting with "Analysis of various deletion constructs in p12..." is elusive.

Page 8.

Second line. "Why "to ensure?"

Line four, Why "co-migrated?"

Edit the last sentence of the paragraph with the omission of "such a function."

Starting with "Corroborating..." - how a motif can be "interaction domain?"

Page 9.

In my file, there is a wrong alignment of numbers and lanes in Fig. 5A.

Second paragraph - assays, not assay.

Page 10.

Repeated "trails?"

Discussion.

The second sentence is difficult to understand, mice humans and yeast are mixed up.

At the end of the second paragraph, there is a mistake - pol zeta consists of the four subunits PMID:25737057.

Figures.

Fig 1B. What asterisk mean?

Fig. 5A. Check alignment.

Supporting information.

Suppl Fig 1. Something is wrong with Western with anti-p50.

Reviewer #3 (Comments to the Authors (Required)):

In this work, Khandagale et al. aim at further describing human DNA polymerase delta, and particularly the properties of its p12 subunit. They identify two important motifs in p12, one RKR motif that they clearly show to be required for p12 subunit dimerization, and one PIP motif that is required for binding to PCNA. They show the same property of *S. pombe* cdm1 subunit that is the homolog of human p12 subunit. They also propose that Pol delta exists under a pentameric

structure composed of p125, p50, p68 and two p12. They conclude from their data that p12 dimerization is required for its interaction with PCNA, and that the pentameric structure they describe is the "native form" of Pol delta.

The data provided in the paper clearly support that p12 is able to dimerize, and that the RKR domain is required for this dimerization. They clearly show also that Cdm1 from *S. pombe* contains a similar motif also required for dimerization. The authors show that RKR domain is also required for PCNA binding. They also clearly evidence the requirement of the PIP motif for the binding of p12 to PCNA.

Besides these two major points, I feel that many conclusions are a little bit overstated...

{section sign}2 and Fig2B: the fact that GFP-p12 and RFP-p12 co-localize doesn't necessarily mean that the replicase contains a dimeric p12. It could as well be several Pol delta in the same focus, some of them containing the GFP-p12, some other the RFP-p12.

{section sign}6 and Fig6: The authors use gel filtration to characterize the sub-assembly of Pol delta. They observe that untagged p12 can exchange with flag-p12 and conclude from this experiment that pol delta contains 2 subunits of p12 and therefore arrange as a pentameric complex. I do not see how this experiment proves that p12 is a dimer in the complex, nor that Pol delta is a pentamer. Either mass spectrometry or sedimentation assays could show the molecular weight of the complex and prove that there are 2 p12 subunits.

{section sign}7 and Fig7: while it is clear that in the absence of the RKR motif, p12 does not dimerize, it is less clear that it is the lack of dimerization that prevents the binding of PCNA. Could the authors rule out that RKR motif is not needed for direct PCNA interaction?

The first part of the discussion also contains many overstatements that should be attenuated. For instance "mutational analyses and several physio-biochemical assays including formaldehyde crosslinking clearly demonstrate that p12 exists as a homodimer both in vivo and in vitro": it is demonstrated in vitro, but it is clearly not demonstrated in vivo. Only co-localizations are shown and could be from different polymerases.

"in vitro reconstitution of Pol-delta5 definitely authenticates our prediction": definitely authenticate is very strong while I do not feel that the authors provide absolute proof of this pentameric assembly.

"our study reveals that the role of RKR-motif in PCNA interaction is mostly indirect": as discussed above, there is no proof that the interaction is not direct through the RKR motif.

While the authors state that the "pentameric complex is the native form" of Pol delta, they then say that "dimerization could be another mode of regulation".

From that point, the discussion becomes very hypothetical: they mention the proteolysis of p12 and p68 by human calpain-1 that could regulate or modulate the composition of Pol delta. But there is no reference. Has this been described before, or is this statement completely hypothetical?

"Pol delta4 with a monomeric p12 does not exist in the cell": again a very strong overstatement. How was tetrameric Pol delta was purified if it doesn't exist? Is their putative pentameric protein functional?

The final model (and figure 10) is also very hypothetical suggesting p12 proteasomal degradation as a response to DNA damage, or cleavage of p12 and p68 in "certain situations". This should be clarified with references.

Overall, the description of p12 RKR and PIP domains, and the dimerization of the subunit are convincing. The rest of the points discussed in the present paper should be softened or consolidated with experiments.

I also feel that the manuscript needs some serious rewriting and English editing. Some sentences are badly written to the point that they are very difficult to understand.

Other points:

- abstract: "three DNA transaction processes": which are they?
- abstract: IDCL: should be defined
- introduction: DNA replication doesn't "suppress" mutagenesis
- introduction: Pol alpha is not a replicase
- {section sign}1: "and such interactions are not mediated through any yeast proteins": how could the author exclude this?
- {section sign}5: CD should be defined.
- many figures are referenced improperly in the text.

Editor, LSA

Feb. 26th, 2019

Dear Editor,

Attached, please find our revised version of the manuscript (LSA-2019-00323-T) entitled “Human DNA polymerase delta is a pentameric holoenzyme with dimeric p12 subunit” for your consideration for publication in Life Science Alliance. We are grateful to the referees for their critical reading of our manuscript and providing mostly constructive comments. Kindly, find the detailed point-by-point response to the reviewer’s comments.

Reviewer #1:

The manuscript shows a range of experiments that, taken together, provide convincing evidence that p12 subunit of the Pol Delta holoenzyme is dimeric in solution and when integrated into the holoenzyme. This is an important point that the field should be aware of. Interestingly, they provide evidence that the dimer interaction is required for the ability of p12 to interact with PCNA and map for the first time the PCNA interacting region. The data and information in the manuscript is definitely publishable and advances our understanding of the Pol Delta complex. There are a number of issues that the authors need to clarify before publication and the manuscript should be shortened.

Reply: We thank the reviewer for appreciating our multi-experimental approach to decipher dimerization of p12 subunit both in solution and in holoenzyme, and supporting publication in LSA. Discussion portion is now shortened.

Page 1. first para. The authors should clarify that pol Epsilon performs leading strand synthesis and Pol delta performs lagging strand synthesis. This has been shown in both yeasts using physical assays. Pol Delta is capable of performing leading strand synthesis when pol Epsilon is absent, and does so in SV40 replication because the CMG helicase, which anchors pol epsilon to the replisome, is not the replicative helicase in these circumstances.

Reply: We agree with the suggestion, and the text is now modified keeping many recent reviews in consideration.

General: The authors discuss two yeasts. In some places they say "in yeast". It needs to be specified which yeast. The language could do with a further edit.

Reply: Thank you for pointing out. We have now added *S. cerevisiae* or *S. pombe* in relevant position of the text. We have taken utmost care to improve the readability of our paper.

Many figures could be combined to generate a 4 or 5 figures manuscript.

Reply: It's a good idea, but we hope that the reviewer will recognize the merit of our figure positioning in the context of different sections of the result.

Page 6 para 2 (relating to figure 2). The immunofluorescence data do not distinguish if the GFP and RFP p12 tagged complexes are in the same pol delta complex or different pol delta complexes that are co-located in the same replication factory. These data are therefore over-interpreted in the manuscript. The authors cannot say these data suggest that both proteins are part of a replication unit and function together as an oligomeric protein.

Reply: Yes, we agree with the reviewer and it's quite possible that multiple Pol δ enzymes could exist in a single foci. However, our conclusion is based on the control experiment. Various reports have demonstrated that DNA polymerases like Pol eta, Pol iota, Pol kappa and others apart from replicative polymerases interact with PCNA and develop discrete foci in human cell lines. Here, DNA polymerase theta was used as a negative control which also forms discrete foci but did not co-localize with p12, even though both Pol θ and p12 interact with PCNA, By reviewer's argument, we should have noticed yellow foci, but we did not. Therefore, our argument has merit to suggest oligomerization of p12. However, we have softened the conclusion and mentioned that "We suggest from these observations that p12 could function in replication factories as a potential oligomeric protein with PCNA".

Page 7 para 1. Figure 3C is referred to as figure 4C

Reply: Corrected.

Page 10 fist line of para 2. involved should be changed to: required for

Reply: Text is now modified as per the suggestion.

Figure 1 (also Fig 9 B) B. Was reducing agent used (if not, justify, i.e. are there any cystines in p12 (or Cdm1))

Reply: We understand the reviewer's point here. However, by default all our proteins contain 5mM DTT as a part of the storage buffer (already mentioned in materials and methods section) and unlikely dimerization of p12/Cdm1 is because of Disulfide bridge. As can be seen from the amino acid alignment, there are two cysteine residues in human p12 (C₆₁ and C₉₉) away from RKR motif and only one C₈₄ in Cdm1.

Figure 2A. Add sizes for all the bands shown

Reply: For better understanding of the figure, both the panels are now kept in close proximity and have same labeling. The subunits of hPol δ were named as per their molecular sizes.

Figure 2A C. It is unclear what is being measured. Usually ITC injects concentrated protein into a dilute solution. The p12 in the syringe is presumably concentrated (this is not discussed sufficiently in the materials and methods, where more detail for this particular experiment should be given). Thus largely dimeric protein is being injected?. What is producing the heat change? I would have expected to see a plot of the "heat of dilution". i.e. p12 injected into empty buffer. At a minimum this must be discussed as a caveat.

Reply: ITC experiment figures are Fig. 1C and Fig. 7C. We apologize for the oversight. Experimental detail is now added comprehensively in materials and methods, and buffer control experiment is now added to the supplementary section (Supplementary Fig. S4). Maximum average heat change of -0.05 μ cal/s was noticed in buffer control experiments; whereas in experimental heat change was -1.4 to -1 μ cal/s.

Fig3. A it would be helpful to put the mammalian only consensus above the alignment.

Reply: Since the study also deals with *S. pombe* p12 homologue (Cdm1) and functional analysis of conserved RKR- motif, we decided to retain as it is.

Figure 7C. Again, heat of dilution?

Reply: Same explanation as given for **Figure 1 C**

Reviewer #2:

The manuscript describes an impressive set of biochemical, biophysical and in silico experiments documenting that human and fission yeast DNA polymerase delta (pold) is a pentamer with a dimeric smallest subunit, p12. The authors mapped amino acids in N-terminus responsible for dimerization and described how alteration of the motif affects the interaction of p12 with PCNA via PIP motif at C-terminus. The observations are new and add to our knowledge of the critical enzyme replicating the human genome. The weakness of the work is the complete absence of functional analyses (i.e., DNA polymerase activity and processivity) of pold with the variants of p12. The discussion could be more concise. We also noticed that the manuscript needs careful editing to avoid typos, jargon (e.g., strains or cell lines are transformed, not plasmids "are transformed") or clumsy wording.

Detailed comments are below.

Abstract.

The equivocal note about "all three DNA transactions" could be better substituted by the concrete "replication, repair and recombination." The sentence starting from "Our mutational..." contains unexplained abbreviations and thus could be understood only by narrow specialists in the field. Typo "dimerzes".

Reply: We thank the reviewer for appreciating our efforts of determining dimerization of p12 subunit of Pol δ in human and *S. pombe* systems; and further its role in PCNA interaction. We agree with the reviewer that in the present study we have not provided any processivity assay of Pol δ complex with p12 mutants. As we have already mentioned in the discussion section, to understand the precise role of various subunits of Pol δ in its processivity, it requires mapping of pip in each subunit and purification of various complexes in different combinations similar to our earlier study in *S. cerevisiae* (Acharya N. et al., PNAS, 2011). In this study, importance has been given to the dimerization property of p12 in Pol δ complex and its subsequent interaction with PCNA. Also, pip of p125 has not been identified yet, hopefully, we will be able to answer those questions in our follow up study. The discussion part is now shortened. As suggested, necessary text modification has been done in the abstract and other sections.

Introduction.

Page 3. In the first sentence, it is better to write "for lowering of the rate of spontaneous mutations and suppressing carcinogenesis." In the second sentence, we suggest adding references to recent reviews PMID:28301743 and PMID:28749073. In the sentence that starts with "Pold not only can..." we suggest to consider addition of the reference to the recent relevant study PMID:30451148. SV40 replication system exploits two pols, not "suggests requirement." For the sentence starting from "Irrespective..." a more focused reference could be found, e.g., PMID:19296856. The last sentence of the first paragraph is unclear. Also, some subunits share a partial similarity, for example, dpb3 and dpb4 possess a histone fold. The first sentence of the second paragraph might be a stretch. Can the authors rephrase "the deletion is not achievable?" What is a "genotoxic burden"?

Page 4. Proofreading, not "proof reading." The statement in the sentence starting with "Except in p68 and p50..." does not take into account the paper PMID:22119860. "Therefore" in the last paragraph could be omitted.

Reply: As suggested text has been rectified and new references are now included. "the deletion is not achievable?" has now been modified to "Cdc27 deletion strain of *S. pombe* is not viable" and "genotoxic burden" is now changed to "DNA damage" for clarity. Space in "proof reading" is removed.

Results.

Page 4. The second sentence should be edited. The third should be edited too. It is incorrect to say that a subunit "was more than a 1:1 ratio." Can the authors comment of a relatively high abundance of p68?

Reply: Both the sentences are now edited. Being p12 the smallest subunit, higher stoichiometry of this protein in various preparations of Pol δ is quite apparent and consistently noticed. However, other subunits including p68 exist mostly in equimolar stoichiometry.

Page 5. Why "co-transformed" in the third sentence? Did the authors try a second marker (lacZ) for the two-hybrid system? The last sentence of the top paragraph is not clear. "Closed" or "close" in the third paragraph? How one can "decipher" oligomerization? What is "irrespective of any antibody?"

Reply: Thank you for pointing out, "co-transformed" is rectified. We did not estimate the β -galactosidase activity rather opted for auxotrophic marker selection in our yeast two hybrid assay. The last sentence is now removed as suggested by other reviewer as well. "Closed to" is now edited to "approximately"; and "decipher" is replaced with "establish". The sentence "irrespective of any antibody" is now modified for better comprehension.

Page 6. Second sentence - more explanations of Fig 2A is desirable, the results look similar for both pull-downs. Second paragraph. How p68 and 50 were "deciphered? " What is the rationale to use pol theta as a control?

Reply: The pull down results will be same only when both the antibodies will precipitate the native Pol δ . The result is convincing because we could detect more GFP-p12 signal when anti-GFP antibody was used, similarly more of p125 band in the precipitate due to immune-precipitation by anti-p125-antibody was observed. Extra text has been added to this section for better understanding. Similar to this study, fluorescence based co-localization of p68 and p50 subunits with PCNA were studied. Here DNA polymerase theta has been used as a negative control and its application to the study has been explained as raised by reviewer #1 and #3.

Page 7. The meaning of the sentence starting with "Analysis of various deletion constructs in p12... " is elusive.

Reply: Modified and unrelated sentences are removed.

Page 8. Second line. "Why "to ensure?" Line four, Why "co-migrated?" Edit the last sentence of the paragraph with the omission of "such a function." Starting with "Corroborating..." - how a motif can be "interaction domain?"

Reply: Sentences are now modified.

Page 9. In my file, there is a wrong alignment of numbers and lanes in Fig. 5A. Second paragraph - assays, not assay.

Reply: Corrected.

Page 10. Repeated "trails?"

Reply: Corrected.

Discussion.

The second sentence is difficult to understand, mice humans and yeast are mixed up. At the end of the second paragraph, there is a mistake - pol zeta consists of the four subunits PMID:25737057.

Reply: The statement is now corrected.

Figures.

Fig 1B. What asterisk mean?

Reply: * indicates degraded protein of Carbonic anhydrase.

Fig. 5A. Check alignment.

Reply: Corrected.

Supporting information. Suppl Fig 1. Something is wrong with Western with anti-p50.

Reply: The data is not adding any more information, so we have deleted the panel.

Reviewer #3:

In this work, Khandagale et al. aim at further describing human DNA polymerase delta, and particularly the properties of its p12 subunit. They identify two important motifs in p12, one RKR motif that they clearly show to be required for p12 subunit dimerization, and one PIP motif that is required for binding to PCNA. They show the same property of S. pombe cdm1 subunit that is the homolog of human p12 subunit. They also propose that Pol delta exists under a pentameric

structure composed of p125, p50, p68 and two p12. They conclude from their data that p12 dimerization is required for its interaction with PCNA, and that the pentameric structure they describe is the "native form" of Pol delta. The data provided in the paper clearly support that p12 is able to dimerize, and that the RKR domain is required for this dimerization. They clearly show also that Cdm1 from *S. pombe* contains a similar motif also required for dimerization. The authors show that RKR domain is also required for PCNA binding. They also clearly evidence the requirement of the PIP motif for the binding of p12 to PCNA. Besides these two major points, I feel that many conclusions are a little bit overstated.

{section sign}2 and Fig2B: the fact that GFP-p12 and RFP-p12 co-localize doesn't necessarily mean that the replicase contains a dimeric p12. It could as well be several Pol delta in the same focus, some of them containing the GFP-p12, some other the RFP-p12.

Reply: Yes, it could be possible. However, the negative control used in this experiment supports our argument. We have already explained in detail as raised by reviewer#1. Generally, co-localization analysis of various proteins in replicating foci is a well-established technique to determine protein-protein interaction in the nucleus; provided the result is further supported by biochemical techniques. Since, p12 interacts with itself in *in vivo* context as determined by yeast two hybrid analysis, we believe our argument has merit. Nevertheless, we have modified our conclusion for this assay as follows "We suggest from these observations that p12 could function in replication factories as a potential oligomeric protein of Pol δ with PCNA".

{section sign}6 and Fig6: The authors use gel filtration to characterize the sub-assembly of Pol delta. They observe that untagged p12 can exchange with flag-p12 and conclude from this experiment that pol delta contains 2 subunits of p12 and therefore arrange as a pentameric complex. I do not see how this experiment proves that p12 is a dimer in the complex, nor that Pol delta is a pentamer. Either mass spectrometry or sedimentation assays could show the molecular weight of the complex and prove that there are 2 p12 subunits.

Reply: Respectfully, we don't agree with the reviewer's comment on pentameric nature of Pol δ . As per the current notion in the field, human Pol δ is a tetrameric holoenzyme. To the best of our knowledge, neither mass spectrometry nor sedimentation assay has been carried out to confirm the molecular size of this complex. However, size exclusion chromatography has allowed to purify Pol δ complex with all the subunits in the *in vitro* reconstitution experiment. By using similar approach we are able to detect two population of Pol δ complexes (i) Pol δ with FLAG-p12 and (ii) Pol δ with FLAG- p12 and untagged p12.

Five bands of Pol δ are easily detectable in the second population. Since, 1st population of complex elutes prior to 2nd population Pol δ in gel filtration, it convincingly suggests that the molecular size 1st complex is bigger than the 2nd population. Thus, both the complexes are pentameric in nature. As we don't get enough of these complexes, at present it is not feasible to carry out SAXS or analytical ultracentrifugation for further validation.

{section sign}7 and Fig7: while it is clear that in the absence of the RKR motif, p12 does not dimerize, it is less clear that it is the lack of dimerization that prevents the binding of PCNA. Could the authors rule out that RKR motif is not needed for direct PCNA interaction?

Reply: Yes, we agree that it is a tricky question to answer unless we get some structural evidence. ITC in fig. 7C ii confirms this to some extent. P12 pip motif mutant that is capable of forming a dimer failed to bind PCNA rules out RKR's direct role in PCNA binding (Fig. 7).

The first part of the discussion also contains many overstatements that should be attenuated. For instance "mutational analyses and several physio-biochemical assays including formaldehyde crosslinking clearly demonstrate that p12 exists as a homodimer both in vivo and in vitro": it is demonstrated in vitro, but it is clearly not demonstrated in vivo. Only co-localizations are shown and could be from different polymerases. "in vitro reconstitution of Pol-delta5 definitely authenticates our prediction": definitely authenticate is very strong while I do not feel that the authors provide absolute proof of this pentameric assembly.

Reply: The discussion part is now shortened and some of these points are now explained. Oligomerization of a given protein can be verified in the cell as carried out in this study for p12 by yeast two hybrid, co-localization and immunoprecipitation assays, which was further confirmed by biochemical techniques to confirm dimeric nature of p12. Explanation for p12 subunits co-localization and purification of Pol δ 5 has been already given for the conclusion that we have drawn for the study. We are not sure of feasibility of verifying dimerization but not oligomerization of a protein in the cell other than methods used in the study. As pointed out by the reviewer we have now modified some of these strong statements.

"our study reveals that the role of RKR-motif in PCNA interaction is mostly indirect": as discussed above, there is no proof that the interaction is not direct through the RKR motif.

Reply: Explanation has been given.

While the authors state that the "pentameric complex is the native form" of Pol delta, they then say that "dimerization could be another mode of regulation".

Reply: Thanks for pointing out and we understand the confusion. We have now rectified the statement.

From that point, the discussion becomes very hypothetical: they mention the proteolysis of p12 and p68 by human calpain-1 that could regulate or modulate the composition of Pol delta. But there is no reference. Has this been described before, or is this statement completely hypothetical?

Reply: It is not hypothetical. References (14, 25, 26) were added that talked about degradation of p12 and p68.

"Pol delta4 with a monomeric p12 does not exist in the cell": again a very strong overstatement. How was tetrameric Pol delta was purified if it doesn't exist? Is their putative pentameric protein functional?

Reply: As explained earlier for the *in vitro* reconstitution of pentameric Pol δ section, earlier characterized Pold4 is infact Pold5. Purification of earlier reported Pol δ 4 does not mean that it has monomeric p12. Somehow the dimeric p12 in those Pol δ was overlooked by other groups.

The final model (and figure 10) is also very hypothetical suggesting p12 proteasomal degradation as a response to DNA damage, or cleavage of p12 and p68 in "certain situations". This should be clarified with references.

Reply: Fig. 10 A is not hypothetical and supported by references (14, 25, 26). However Fig. 10 B is hypothetical and requires validation.

Overall, the description of p12 RKR and PIP domains, and the dimerization of the subunit are convincing. The rest of the points discussed in the present paper should be softened or consolidated with experiments. I also feel that the manuscript needs some serious rewriting and English editing. Some sentences are badly written to the point that they are very difficult to understand.

Reply: Thank you for appreciating our work. We hope the reviewer will be satisfied with the explanations provided here. As pointed out many of the strong statements were now softened. we have now taken maximum care to improve the readability of the manuscript. Thanks.

Other points:

-abstract: "three DNA transaction processes": which are they?

Reply: Now it is revised.

-abstract: IDCL: should be defined

Reply: It is now defined.

-introduction: DNA replication doesn't "suppress" mutagenesis

Reply: It is revised as suggested by earlier reviewer.

-introduction: Pol alpha is not a replicase

Reply: The statement is modified.

-{section sign}1: "and such interactions are not mediated through any yeast proteins": how could the author exclude this?

Reply: excluded. Thanks.

-{section sign}5: CD should be defined.

Reply: Thanks, modified.

-many figures are referenced improperly in the text.

Reply: Thanks for pointing out. We have now corrected figure numbers.

Once again, we would like to thank the Reviewers and Editor for their thoughtful comments and hope that the improved version of the manuscript will be suitable for acceptance.

Sincerely yours,

Narottam Acharya

February 27, 2019

RE: Life Science Alliance Manuscript #LSA-2019-00323-TR

Dr. Narottam Acharya
Institute of Life Sciences
Infectious Disease Biology
Nalco Square
Bhubaneswar, Odisha 751023
India

Dear Dr. Acharya,

Thank you for submitting your revised manuscript entitled "Human DNA polymerase delta is a pentameric holoenzyme with dimeric p12 subunit". We appreciate the introduced changes and would be happy to publish your paper in Life Science Alliance pending final minor revisions:

- The language needs further editing, please do. I copy an already edited abstract below as guidance.
- The blots in figure 2 and 6 are overcontrasted, please provide better ones as well as the source data for these blots.
- All individual figures need to be on a single page, please re-arrange those that go over two pages at the moment (Figures 1, 3, 7).
- Please list 10 authors et al. in the reference list.
- Please mention the number of replicates performed for each experiment.
- Please add a scale bar in Figure 2.

A. FINAL FILES:

B. MANUSCRIPT ORGANIZATION AND FORMATTING:

Sincerely,

Human DNA polymerase delta (Pol δ), a holoenzyme consisting of p125, p50, p68 and p12 subunits, plays an essential role in DNA replication, repair and recombination. Herein, using multiple physicochemical and cellular approaches we found that the p12 protein forms a dimer in solution. In vitro reconstitution and pull-down of cellular Pol δ by tagged p12 substantiates the pentameric nature of this critical holoenzyme. Further, a consensus PCNA interaction protein motif at the extreme carboxyl terminal tail and a homodimerization domain at the amino-terminus of the p12 subunit were identified. Mutational analyses of these motifs in p12 suggest that dimerization facilitates p12 binding to the inter-domain connecting loop of PCNA. Additionally, we observed that oligomerization of the smallest subunit of Pol δ is evolutionarily conserved as Cdm1 of *S. pombe* also dimerizes. Thus, we suggest that hPol δ is a pentameric complex with a dimeric p12 subunit and we discuss implications of p12 dimerization on enzyme architecture and PCNA interaction during DNA replication.

Editor, LSA

March 4th, 2019

Dear Editor,

Attached, please find our minor revised version of the manuscript (LSA-2019-00323-T-R) entitled "Human DNA polymerase delta is a pentameric holoenzyme with a dimeric p12 subunit" for your consideration for publication in Life Science Alliance. We thank the reviewers and the editorial team for accepting our manuscript for publication in LSA. Kindly, find the detailed point-by-point response to the editorial comments.

Editorial Comments

Thank you for submitting your revised manuscript entitled "Human DNA polymerase delta is a pentameric holoenzyme with a dimeric p12 subunit". We appreciate the introduced changes and would be happy to publish your paper in Life Science Alliance pending final minor revisions:

- The language needs further editing, please do. I copy an already edited abstract below as guidance.

Reply: Thanks a lot. We have now edited and hope you will appreciate our efforts.

- The blots in figure 2 and 6 are overcontrasted, please provide better ones as well as the source data for these blots.

Reply: Corrected.

- All individual figures need to be on a single page, please re-arrange those that go over two pages at the moment (Figures 1, 3, 7).

Reply: Now all the figures are in one page.

- Please list 10 authors et al. in the reference list.

Reply: Corrected.

- Please mention the number of replicates performed for each experiment.

Reply: Mentioned in the materials and method section for the respective experiments.

- Please add a scale bar in Figure 2.

Reply: Scale bar is now added in the figure and it is defined in the figure legend.

Once again, we would like to thank the Reviewers and Editor; and hope that the improved version of the manuscript will be suitable for publication.

Sincerely yours,

Narottam Acharya

March 11, 2019

RE: Life Science Alliance Manuscript #LSA-2019-00323-TRR

Dr. Narottam Acharya
Institute of Life Sciences
Infectious Disease Biology
Nalco Square
Bhubaneswar, Odisha 751023
India

Dear Dr. Acharya,

Thank you for submitting your Research Article entitled "Human DNA polymerase delta is a pentameric holoenzyme with a dimeric p12 subunit". We appreciate the introduced changes and it is a pleasure to let you know that your manuscript is now accepted for publication in Life Science Alliance. The original source data provided for Figures 2 and 6 (Bio-Rad (TM) files and tif versions thereof) will be uploaded to the HTML version of your paper. Note that the resolution and contrast is still not ideal for publication.

*****IMPORTANT:** If you will be unreachable at any time, please provide us with the email address of an alternate author. Failure to respond to routine queries may lead to unavoidable delays in publication.*******

DISTRIBUTION OF MATERIALS:

Again, congratulations on a very nice paper. I hope you found the review process to be constructive

and are pleased with how the manuscript was handled editorially. We look forward to future exciting submissions from your lab.

Sincerely,
